# Ran pathway-independent regulation of mitotic Golgi disassembly by Importin-α

Chih-Chia Chang[1], Ching-Jou Chen[2], Cédric Grauffel[3], Yu-Chung Pien[2], Carmay Lim [3], Su-Yi Tsai[2,4] & Kuo-Chiang Hsia [1,5]

To facilitate proper mitotic cell partitioning, the Golgi disassembles by suppressing vesicle fusion. However, the underlying mechanism has not been characterized previously. Here, we report a Ran pathway-independent attenuation mechanism that allows Importin-α (a nuclear transport factor) to suppress the vesicle fusion mediated by p115 (a vesicular tethering factor) and is required for mitotic Golgi disassembly. We demonstrate that Importin-α directly competes with p115 for interaction with the Golgi protein GM130. This interaction, promoted by a phosphate moiety on GM130, is independent of Importin-β and Ran. A GM130 K34A mutant, in which the Importin-α-GM130 interaction is specifically disrupted, exhibited abundant Golgi puncta during metaphase. Importantly, a mutant showing enhanced p115-GM130 interaction presented proliferative defects and G2/M arrest, demonstrating that Importin-α-GM130 binding modulates the Golgi disassembly that governs mitotic progression. Our findings illuminate that the Ran and kinase-phosphatase pathways regulate multiple aspects of mitosis coordinated by Importin-α (e.g. spindle assembly, Golgi disassembly).

[1] Institute of Molecular Biology, Academia Sinica, Taipei 11529, Taiwan. [2] Department of Life Science, National Taiwan University, Taipei 10617, Taiwan. [3] Institute of Biomedical Sciences, Academia Sinica, Taipei 11529, Taiwan. [4] Genome and Systems Biology Degree Program, National Taiwan University and Academia Sinica, Taipei 10617, Taiwan. [5] Institute of Biochemistry and Molecular Biology, College of Life Sciences, National Yang-Ming University, Taipei 11221, Taiwan. Correspondence and requests for materials should be addressed to S.-Y.T. (email: suyitsai@ntu.edu.tw) or to K.-C.H. (email: khsia@gate.sinica.edu.tw)

During interphase, the Golgi apparatus of the cytoplasmic endomembrane system carries out critical functions in the protein secretory pathway, such as receiving transport vesicles from the endoplasmic reticulum (ER) and directing them to their destinations[1,2]. The vesicle tethering factor p115 and the Golgi peripheral membrane protein GM130 facilitate targeting of transport vesicles to the Golgi[3,4]. However, during mitosis, p115-mediated vesicle fusion to Golgi membranes is reduced, thereby down-regulating the secretory pathway[5–7] and triggering the Golgi vesiculation necessary for Golgi partitioning during cell division[8]. Mitotic phosphorylation of GM130 on serine 25 (Ser-25) has been proposed to account for the reduced binding of p115 to Golgi membranes[5–7,9,10], but whether serine phosphorylation alone is solely responsible for inhibiting p115 binding has not been proven empirically.

Recently, an evolutionarily conserved and nuclear localization signal (NLS)-like motif was identified in the N-terminal region of GM130 that binds p115[11–13]. Conventionally, Importin-α-dependent classical NLS (cNLS) includes: (1) a monopartite element that consists of 4–8 positively charged residues and primarily binds to the major binding site of Importin-α (e.g. SV40 large T antigen[14]); and (2) a bipartite element that contains two clusters of basic amino acids separated by a linker region and that binds to the major and minor binding sites of Importin-α (e.g. nucleoplasmin[15,16]). In the nuclear import pathway, cNLS-containing proteins assemble into a ternary complex, with Importin-α and Importin-β in equal stoichiometries, which is transported across the nuclear envelope (NE) through nuclear pore complexes (NPCs) via interaction with nucleoporins that contain phenylalanine–glycine (FG) repeats[17]. Subsequently, in the nucleus, GTP-bound Ran binds to Importin-β and disassembles the ternary complex, releasing the imported cargo proteins[17]. Guanine nucleotide exchange factors (GEFs) (e.g. chromatin-associated protein RCC1) are required to switch Ran from GDP-bound to GTP-bound states and to confine localization of RanGTP in the nucleus[17]. Although the NLS-like motif of GM130 does interact with Importin-α[13], sub-stoichiometries of Importin-α and Importin-β have been associated with Xenopus egg membrane fractions and can be pulled down by GM130 from mitotic cell lysates[13,18]. These observations suggest that binding of Importin-α to GM130 without Importin-β may be carried out by a non-canonical NLS.

Numerous membrane proteins situated in the inner membrane (INM) of the NE are actively transported to the nucleus by the Importin-α/-β-mediated pathway, facilitated by NLS[19]. Structural and biochemical analyses have further suggested that the NLS of the INM proteins Heh1/Heh2 (*S. cerevisiae*) and Pom121 (metazoans) adopts a non-canonical conformation and bears unique biochemical properties, allowing displacement of the N-terminal Importin-β binding domain (IBB) of Importin-α that competes for binding with the NLS-proteins and thereby bypassing auto-inhibition[20,21]. Thus, these INM proteins bind Importin-α in the absence of Importin-β[20,21]. As both Golgi and NE are part of the cellular endomembrane system, it is plausible that the Golgi membrane-associated GM130 protein carries an NLS-like motif that has evolutionarily conserved structural and biochemical properties shared with the NLS of INM proteins.

The NLS-like motif of GM130 was proposed to sequester Importin-α from TPX2, thereby stimulating microtubule nucleation during mitosis[13]. Since both Importin-α and p115 bind to the GM130 N-terminus and Importin-α pulldown by GM130 is enriched during mitosis, we hypothesize that binding of Importin-α to GM130 during mitosis may suppress the p115 and GM130 interaction, contributing to Golgi disassembly. Importantly, this non-canonical interaction may be sustainably resistant to mitotic RanGTP-induced liberation, ensuring Golgi vesiculation.

To test our hypothesis, we first biochemically demonstrate that binding of Importin-α and p115 to GM130 is mutually exclusive. GM130 directly interacts with full-length Importin-α in the absence of Importin-β, which is enhanced by phosphorylation of the Ser-25 residue, and this interaction is resistant to Ran pathway liberation. Next, our crystal structure of the Importin-α and GM130-NLS complex reveals a non-conventional binding mode that accounts for a higher affinity for full-length Importin-α compared to the IBB domain. Moreover, crystal structures of Importin-α with recombinant GM130 either carrying a phosphomimetic residue or a phosphate moiety and molecular dynamic (MD) simulations show that the phosphate moiety indirectly enhances the affinity of Importin-α for GM130 by extending the intermolecular interaction network.

Next, we introduce a single point mutation (K34A) in GM130 to biochemically disrupt its interaction with Importin-α. Interestingly, the heterozygous mutant shows proliferative defects and punctate Golgi during metaphase. Furthermore, in vitro immunoprecipitation and in situ proximity ligation assays (PLA) reveal elevated levels of p115 and GM130 during mitosis in the mutant cell line. Our data describe a molecular mechanism whereby Importin-α modulates mitotic Golgi morphology via a Ran pathway-independent interaction with GM130.

## Results

**Binding of Importin-α suppresses the p115-GM130 interaction.** Phosphorylation of GM130 on Ser-25 is correlated with p115 dissociation, with this process appearing to be specific to mitosis and being responsible for reduction of p115-mediated vesicle fusion to Golgi membranes[5–7,9,10]. The N-terminal region of GM130 and the p115 C-terminus interact[11,12], so we purified recombinant proteins containing these two regions and then assessed their binding affinity by isothermal titration calorimetry (ITC). The wild type GM130 N-terminal region (a.a. 1-85; hereafter GM130-WT) directly bound p115 (a.a. 780–930; hereafter p115-CTD) (Fig. 1a). Furthermore, we examined if GM130 phosphorylation reduces the binding affinity for p115. $K_d$ values determined by ITC were comparable between p115-CTD and GM130-WT or a phosphomimetic GM130 (1–85) mutant in which Ser-25 was substituted by Asp (hereafter GM130-S25D) (Fig. 1a, b). Thus, under our experimental conditions, phosphorylation of GM130 Ser-25 does not substantially perturb the GM130 and p115 interaction. Notably, the N-terminus of GM130 carries a consensus NLS-like sequence that interacts with Importin-α during mitosis[13]. We found in competition-binding assays that p115-CTD pulldown by GST-GM130-S25D was reduced when we added increasing amounts of Importin-α lacking the IBB domain (a.a. 70–524; hereafter Importin-α (ΔIBB)) (Fig. 1c). Additionally, even though the salt concentration used in ITC measurement for Importin-α(ΔIBB) and GM130 was higher, Importin-α(ΔIBB) and GM130 still displayed a greater binding affinity compared to that of p115-CTD and GM130 (200 vs. 50 mM; Fig. 1d, e). Therefore, we propose that binding of Importin-α to GM130 could be primarily responsbile for preventing interaction between p115 and GM130 during mitosis.

**An atypical NLS-binding mode between GM130 and Importin-α.** Next, we sought to determine the crystal structure of the GM130 N-terminal region plus Importin–α(ΔIBB) complex to understand their binding mode. The co-purified GM130 peptide (a.a. 1–48)•Importin-α (a.a. 70–498) complex yielded crystals of the $P2_12_12_1$ orthorhombic space group that diffracted to 2.8 Å (Table 1). The structure was determined by molecular replacement using Importin-α as a search model. Although a GM130

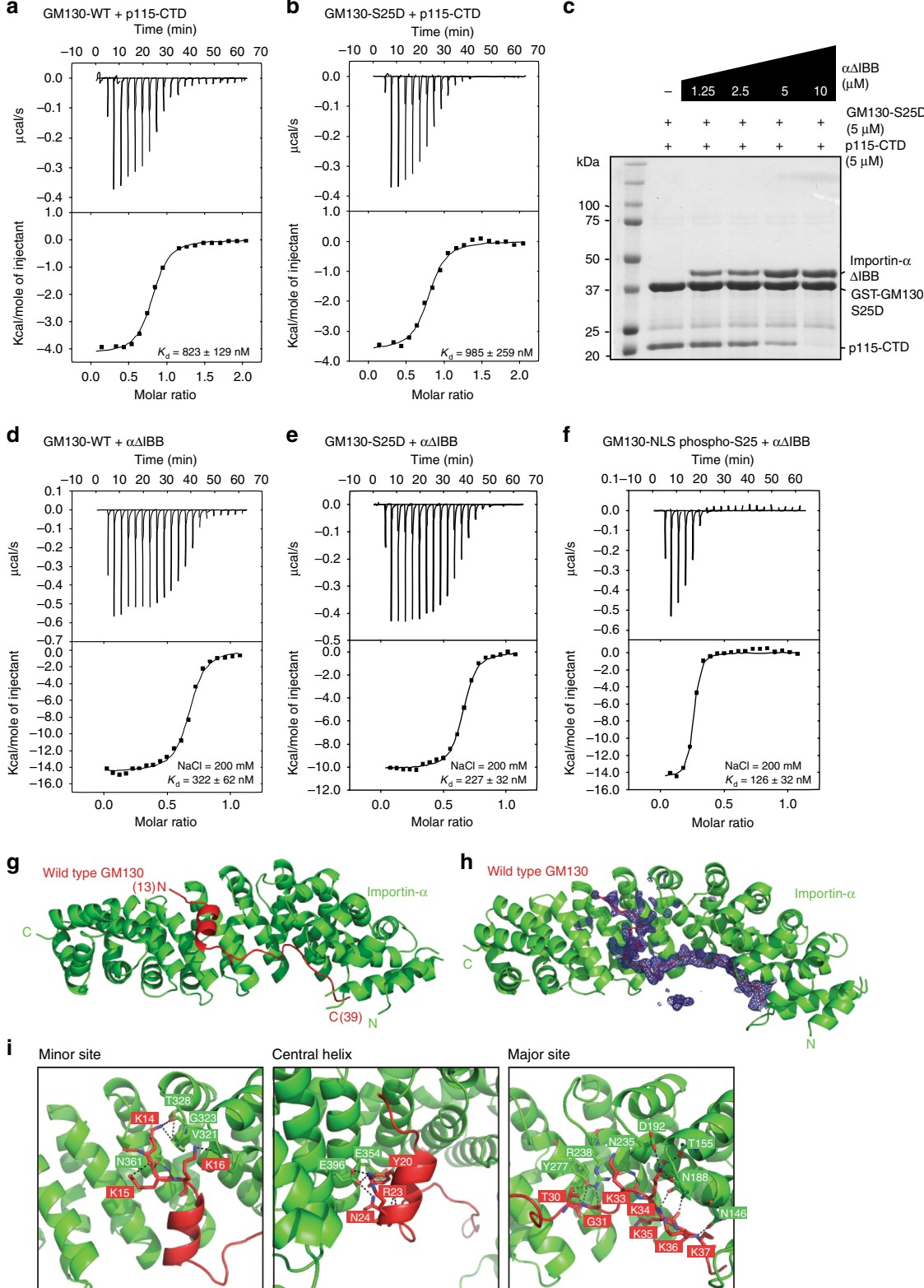

**Fig. 1** Biochemical and structural characterization of the GM130 and Importin-α interaction. **a**, **b** ITC titration curves (upper) and binding isotherms (lower) of p115-CTD with GM130-WT (**a**) and GM130-S25D (**b**). $K_d$ values are indicated. **c** Competition binding assay. GST-GM130-S25D and p115-CTD were incubated with the indicated concentrations of Importin-α(ΔIBB). Samples pulled down by GST beads were analyzed by SDS–PAGE and stained with Coomassie blue. **d**–**f** ITC titration curves (upper) and binding isotherms (lower) of Importin-α(ΔIBB) with GM130-WT (**d**), GM130-S25D (**e**), and GM130-NLS phospho-S25 (**f**). Salt concentration and $K_d$ values are indicated. **g** Cartoon representation of the GM130 (a.a. 1–48)•Importin-α (a.a. 70–498) complex, showing GM130 in red and Importin-α in green. Residues 13–39 in GM130 are assigned. **h** An omit difference (Fo–Fc) map contoured at 2.5 sigma with a superimposed atomic model of the GM130-NLS WT•Importin-α complex. GM130 and Importin-α are displayed as ribbons (red) and cartoons (green), respectively. **i** The three panels highlight the interactions at the minor binding site (left), central α-helix (middle) and major binding site (right)

**Table 1 Data collection and refinement statistics**

|  | GM130-WT | GM130-S25D | GM130-pS25 |
|---|---|---|---|
| *Data collection* |  |  |  |
| Space group | $P2_12_12_1$ | $P2_12_12_1$ | $P2_12_12_1$ |
| Cell dimensions |  |  |  |
| $a, b, c$ (Å) | 75.3, 78.7, 90.2 | 73.7, 79.0, 89.6 | 78.7, 89.8, 99.4 |
| $a, b, g$ (º) | 90.0, 90.0, 90.0 | 90.0, 90.0, 90.0 | 90.0, 90.0, 90.0 |
| Resolution (Å) | 20.0-2.8 (2.90-2.80)[a] | 20.0-1.75 (1.81-1.75)[a] | 20.0-2.4 (2.49-2.40)[a] |
| $R_{sym}$ (%) | 9.7 (54.6) | 6.9 (56.8) | 4.2 (14.2) |
| CC1/2 (%) | 99.6 (90.9) | 99.8 (91.5) | 99.9 (98.3) |
| $\langle I/sI \rangle$ | 16.7 (3.6) | 20.7 (4.3) | 21.5 (7.6) |
| Completeness (%) | 98.7 (91.7) | 95.1 (94.7) | 99.9 (100.0) |
| Redundancy | 5.5 | 7.3 | 4.3 |
| *Refinement* |  |  |  |
| Total reflections | 72,915 | 335,824 | 92,958 |
| Unique reflections | 13,194 | 50,110 | 21,490 |
| $R_{work}/R_{free}$ (%) | 22.4/25.1 | 17.2/19.9 | 17.1/21.6 |
| No. of atoms |  |  |  |
| Protein | 3410 | 3429 | 3483 |
| Water | 12 | 442 | 152 |
| Wilson $B$-factor (Å²) | 72.6 | 27.4 | 39.4 |
| Average $B$-factor (Å²) |  |  |  |
| Importin-a | 86.1 | 37.6 | 55.7 |
| NLS | 81.0 | 36.4 | 104.9 |
| Water | 66.9 | 48.1 | 54.2 |
| Bond length rmsd (Å) | 0.017 | 0.007 | 0.010 |
| Bond angle rmsd (º) | 2.058 | 0.882 | 1.025 |

[a]Values in parentheses are for highest-resolution shell

peptide containing 48 residues was used for co-crystallization with Importin-α (a.a. 70–498), a continuous electron density for the GM130 peptide was clear only from residues 13–39 after initial refinement, so only this portion (referred to as GM130-NLS (a.a. 13–39) hereafter) could be assigned and built unambiguously (Fig. 1g, h).

GM130-NLS (a.a. 13–39) comprised two stretches of basic residues representing the major and minor NLS-binding sites for Importin-α (Fig. 1g, i, Supplementary Fig. 1a, b). Interestingly, we found that a two-turn α-helix that connects these two basic stretches and that is normally disordered in conventional bipartitie NLS is involved in Importin-α recognition and binding (Fig. 1i, Supplementary Fig. 1a). Residues Tyr-20, Gln-22, and Asn-24 within this α-helix are evolutionarily conserved from *Xenopus* to human and provide salt bridges and hydrogen bonds that interact with residues Glu-354, Glu-396, and Lys-353 on Importin-α (Supplementary Fig. 1a, b). Notably, the crystal structure of the native GM130-NLS•Importin-α complex reveals that Ser-25 is located at the bottom of the GM130 α-helix, and is not part of the GM130–Importin-α interaction network (Fig. 2a, b, Supplementary Fig. 1a). However, both the GM130-S25D mutant protein and a synthetic peptide with a phosphate moiety on Ser-25 of GM130 (a.a. 13–40; hereafter GM130-NLS phospho-S25) displayed better binding affinities for Importin-α(ΔIBB) (Fig. 1e, f). Thus, we sought to establish the molecular mechanism by which phosphoserine enhances the affinity of GM130 for Importin-α.

**Ser-25 phosphorylation of GM130 enhances Importin-α binding**. GM130-S25D peptide (a.a. 1–48) or GM130-NLS phospho-S25 was co-purified with Importin-α (70–498) and the protein•peptide complexes were subjected to crystallographic

analysis. Both complexes yielded crystals in the $P2_12_12_1$ space group that diffracted to 1.75 and 2.4 Å respectively (Table 1). Superimposition of GM130-NLS WT and the S25D and phospho-S25 peptides in complex with Importin-α revealed good structural overlay (Fig. 2a, Supplementary Fig. 1c). Interestingly, apart from increased intermolecular interactions in the minor NLS-binding site, we identified intra-molecular interactions involving S25D and phosphoserine in the crystal structures (Fig. 2b–d, Supplementary Fig. 1a): Asp-25 interacts with Gln-21 via water molecules (Fig. 2c), whereas the phosphoserine phosphate interacts with the Arg-18 and Gln-21 side-chains (Fig. 2d).

Next, to analyze whether GM130 phosphorylation endows conformational stability on the GM130•Importin-α complex, thermal-dependent circular dichroism (CD) was performed to determine the melting temperatures ($T_m$) of protein complexes. The temperature scans (wavelength = 222 nm) for Importin-α in complex with GM130-WT, S25D, or phospho-S25 revealed that the phosphorylated protein complexes exhibited increased melting temperatures (Fig. 2e, Supplementary Fig. 1d–f). The $T_m$ was shifted from ~44 ºC for the wild-type complex to ~46 ºC for the phosphorylated protein complexes. The CD, ITC, and crystal structure data give complementary results, suggesting that GM130 phosphorylation indirectly increases the binding affinity of GM130 and Importin-α, leading to enhanced global thermal stability. The overall crystallographic temperature factor ($B$-factor) of the GM130-NLS S25D complex is substantially lower than that of the wild-type complex (Table 1, Supplementary Fig. 2a), also implying that the phosphomimetic mutation increases the stability of the protein complex. Most likely due to different crystallographic packing environments, the GM130-NLS phospho-S25 complex lacked crystal contacts from symmetry-related molecules and displayed a relatively greater $B$-factor, particularly in the central α-helix (Supplementary Fig. 2a, b). However, the highly dynamic α-helix shown in the GM130-NLS phospho-S25 complex indicates that the central α-helix contributes little to binding between Importin-α and GM130 compared to major and minor sites, and few hydrogen bonds are found between the central α-helix and Importin-α (Supplementary Fig. 1a). Thus, as phosphorylated GM130•Importin-α complexes (S25D or phospho-S25) displayed a lower $T_m$ (Fig. 2e), more intermolecular hydrogen bonds (Supplementary Fig. 1a), and reduced $B$-factors (Supplementary Fig. 2a), we propose that phosphorylation of Ser-25 enhances the binding affinity of GM130 for Importin-α via an indirect effect, by blocking the p115 and GM130 interaction (Fig. 2g).

**GM130-NLS phospho-S25 destabilizes the apo state**. To further elucidate how S25 phosphorylation could enhance GM130-NLS −Importin-α binding, we conducted multiple MD simulations of wild-type and phosphorylated GM130-NLS free in solution and in complex with Importin-α. The simulations suggest that destabilization of the phosphorylated peptide relative to the wild-type in solution contributes to the enhanced affinity of phosphorylated GM130-NLS for Importin-α. Whereas the two-turn α-helix was maintained in MD simulations of wild-type and phosphorylated GM130-NLS•Importin-α complexes, the C-terminal region was disrupted in simulations of apo GM130-NLS phospho-S25 (Fig. 2f). This result is due to the S25 phosphate group attempting to form salt bridges with neighboring charged residues, in particular R18 and R23, during the simulations, leading to secondary structure distortion/disruption (Fig. 2g; see the "Discussion" section). When bound to Importin-α, the S25 phosphate enables water-mediated intramolecular interactions with backbone amides of nearby residues. Thus, S25 phosphorylation seems to destabilize the free peptide and stabilize

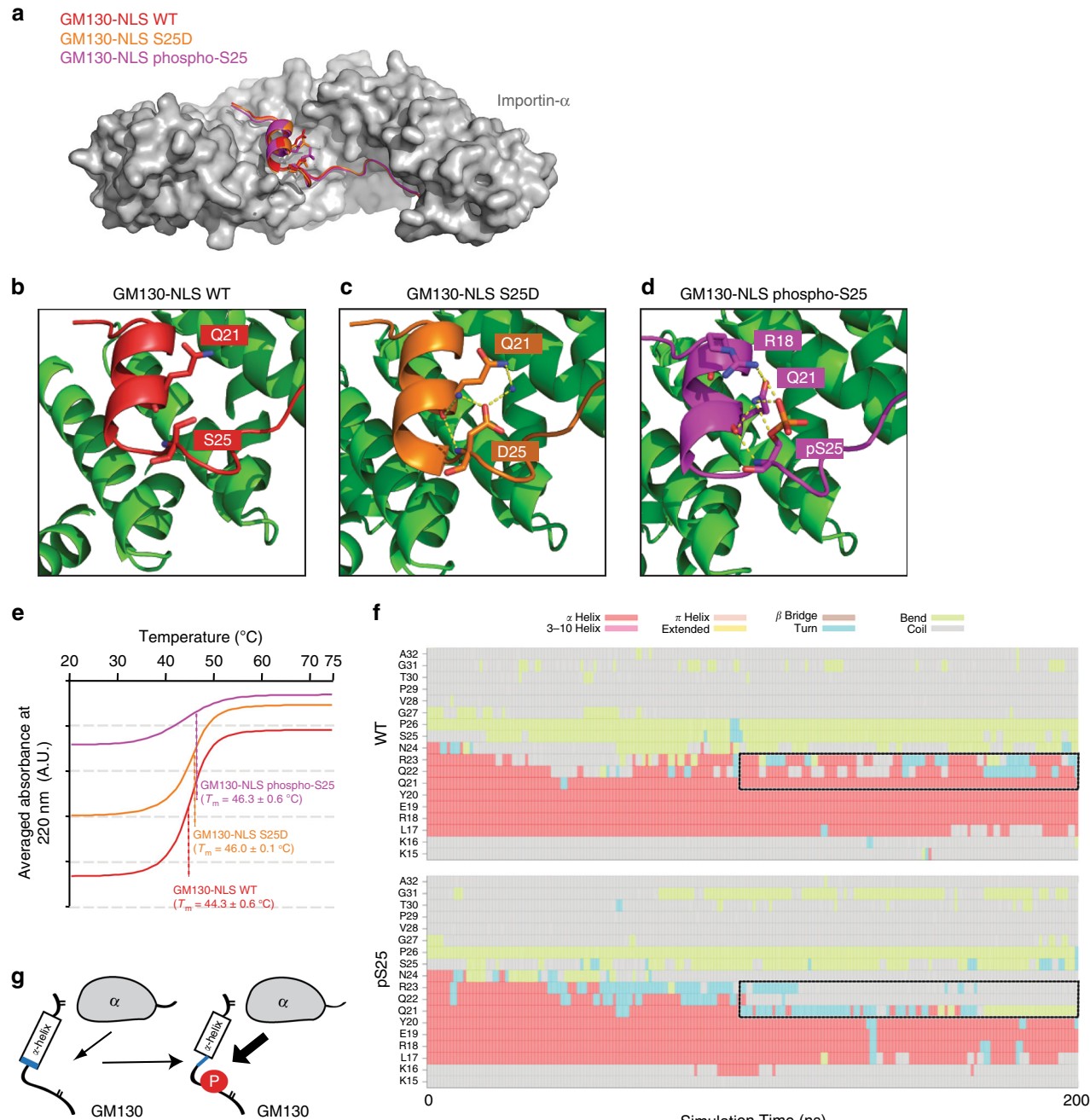

**Fig. 2** A phosphate moiety mediates GM130 intra-molecular interactions. **a** Superimposition of the GM130-NLS WT (red), GM130-NLS S25D (orange) and GM130-NLS phospho-S25 (purple) crystal structures. Importin-α (a.a. 70–498) is shown in gray. **b–d** Detailed views of the residue-residue interactions mediated by Serine-25 (**b**), Aspartic acid-25 (**c**), or Phosphoserine-25 (**d**) in GM130. Water molecules are shown as blue spheres. Hydrogen bonds are indicated by dotted lines. **e** Circular dichroism spectra of temperature scans at a wavelength of 222 nm for GM130-WT (red line), S25D (orange line) or phosphor-S25 (purple line) complexes. Each complex spectrum represents the average of three scans. The melting temperature ($T_m$) of each complex is indicated. **f** Mean secondary structure of GM130-NLS residues 15−32 (averaged over eight 200-ns simulations) in solution as a function of simulation time for native peptide (top) and phosphorylated peptide (bottom). **g** Schematic model of how phosphorylation on GM130-NLS disrupts the α-helix in apo state, enhancing Importin-α binding

the bound form, leading to a slightly higher binding affinity for Importin-α relative to that of the unphosphorylated GM130-NLS peptide.

**Direct interaction between GM130 and full-length Importin-α.** Next, we examined whether GM130-NLS WT or the GM130-NLS phospho-S25 peptide interacts with a higher affinity for full-length Importin-α (hereafter Importin-α FL). Interestingly, similar to what we observed for Importin-α (ΔIBB) (Fig. 1e, f), GM130-NLS phospho-S25 did bind to Importin-α FL with a substantially lower (~3.3-fold) dissociation constant than that of GM130-NLS WT, as revealed by ITC (Fig. 3a, b). Furthermore, recombinant Importin-α FL pulldown by GST-GM130-S25D was also higher relative to that of wild-type (Fig. 3c). We confirmed the interaction of GM130-S25D with Importin-α FL by size-

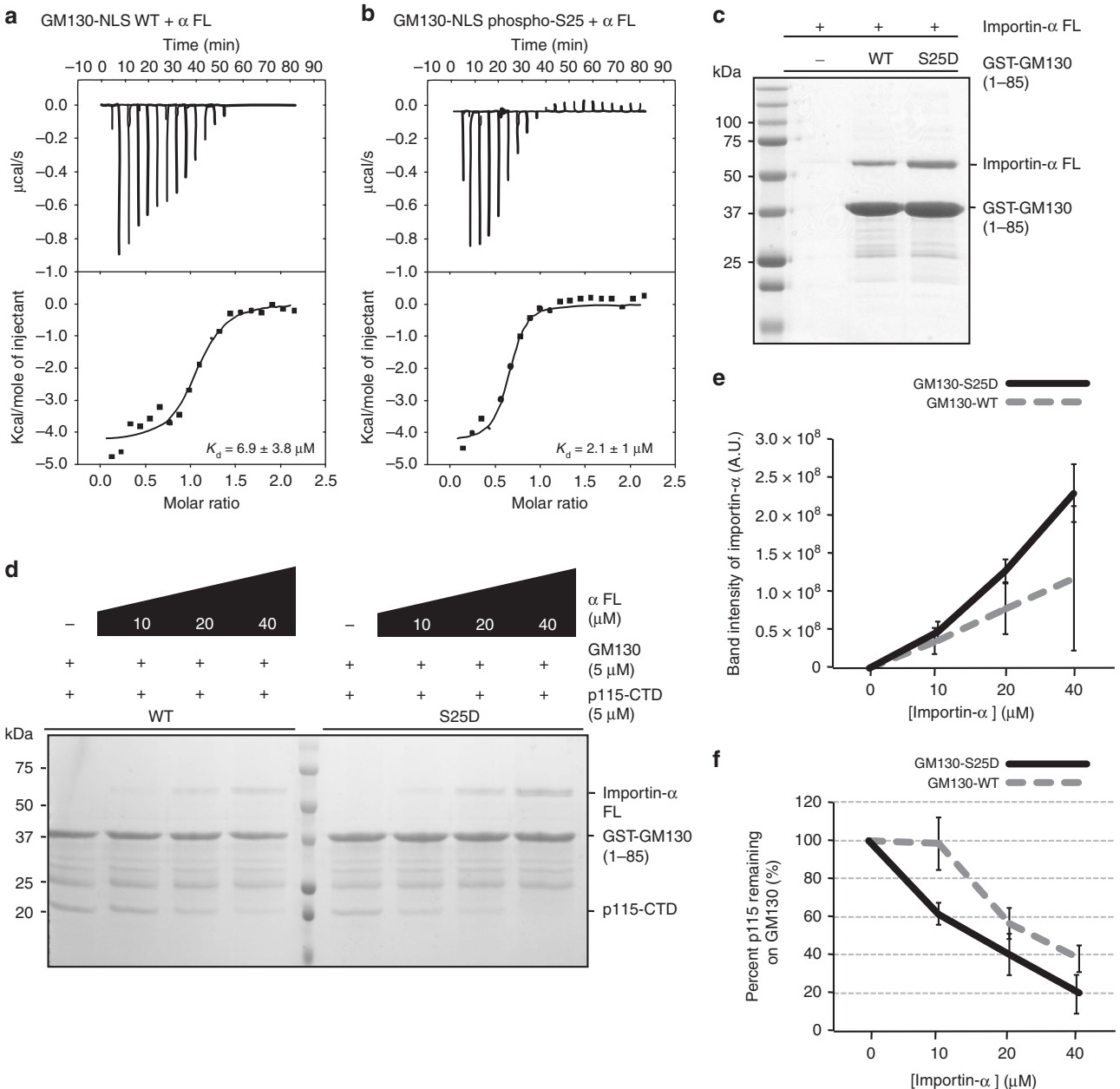

**Fig. 3** GM130 interacts with full-length Importin-α. **a, b** ITC titration curves (upper) and binding isotherms (lower) of Importin-α FL with GM130-NLS WT (**a**) and GM130-NLS phospho-S25 (**b**). $K_d$ values are indicated. **c** Pulldown assays of Importin-α FL with GM130-WT or GM130-S25D. GST-tagged GM-130-WT or GM130-S25D was incubated with Importin-α FL. GST-bound samples were analyzed by SDS–PAGE and stained with Coomassie blue. **d** Competition binding assay. p115-CTD was incubated with either GST-GM130-WT or GST-GM130-S25D in the presence of indicated concentrations of Importin-α FL. Samples pulled down by GST beads were analyzed by SDS–PAGE and stained with Coomassie blue. **e, f** Band intensities of Importin-α FL and p115-CTD from the SDS–PAGE gels in **d** were quantified and normalized by GST-GM130 intensity. Quantitative plots show normalized Importin-α FL intensities (**e**) and percent p115-CTD remaining on GM130 (**f**). Data represent mean ± standard deviation from three independent experiments

exclusion chromatography (SEC) (Supplementary Fig. 3a–d), which revealed a co-migration profile.

To examine whether Importin-α FL also competes with p115-CTD for GM130 binding, we carried out competition binding assays in the presence of Importin-α FL, p115-CTD, and GM130. We found that whereas Importin-α pulldown by GST-GM130-WT was elevated, the percentage of p115 remaining on GM130 was reduced when we added increasing amounts of Importin-α FL (Fig. 3d–f). Hence, as observed for Importin-α (ΔIBB) (Fig. 1c), Importin-α FL competed with p115-CTD for GM130 binding, albeit at a relatively higher concentration of Importin-α FL. Moreover, we observed a ~2-fold increase in Importin-α FL

pulldown by the phosphomimetic mutant (GM130-S25D) compared to wild-type GM130 (40 μM Importin-α FL; Fig. 3d, e). In the presence of 40 μM Importin-α FL, there was about two-fold less p115 remaining on GM130 S25D (~20%) compared to wild-type (~38%) (Fig. 3d, f). These results suggest that the phosphomimetic mutant has a greater affinity toward Importin-α FL than wild-type, so it exhibits better competitive displacement of p115 by Importin-α FL.

We then assessed whether the GM130-S25D phosphomimetic mutant could enhance Importin-α pulldown from interphase mammalian HEK293T cell lysates. Interestingly, Importin-α pulldown from these cell lysates by GST-GM130-S25D was

increased ~1.3-fold compared with that of wild-type (Supplementary Fig. 3e, g), suggesting that GM130 phosphorylation does enhance binding between GM130 and Importin-α FL. Furthermore, under the same acquisition conditions, we detected a more than 10-fold reduced Western blot signal of Importin-β compared to that of Importin-α FL (Supplementary Fig. 3e, g), demonstrating sub-stoichiometric pulldown of Importin-α and -β by GM130. Additionally, probably due to the presence of abundant NLS-containing proteins in the cell lysates, only a small fraction of Importin-α and Importin-β were pulled down by GM130 even though we incubated an excess of GST-GM130 recombinant protein (Supplementary Fig. 3e, f). However, this outcome still implies that phosphorylated GM130 has a better affinity for Importin-α than wild-type GM130, even in the presence of other NLS proteins.

**GM130 bears an IBB-like NLS motif**. We found that: (1) the GM130-NLS displayed a higher binding affinity for Importin-α (ΔIBB) compared to the IBB domain alone (Fig. 1f and Supplementary Fig. 3h; ~126 vs. ~944 nM); (2) IBB was essentially not able to displace GM130 at an 8-fold higher concentration, as demonstrated by in vitro pulldown assays (Supplementary Fig. 3i); and (3) superimposition of Importin-α (ΔIBB)-bound GM130-NLS with an inner nuclear membrane protein (Heh2-NLS[21]) revealed structural similarities (Supplementary Fig. 3j). Therefore, we examined whether GM130-NLS functions as an IBB-like NLS, allowing it to interact with Importin-β. Whereas Heh2-NLS does not bind Importin-β[21], we observed co-migration of GM130-NLS WT and Importin-β by SEC (Fig. 4a–d), suggesting that these latter interact. Furthermore, ITC revealed an exothermic profile with sub-micromolar-binding affinity between GM130-NLS WT and Importin-β (~1.2 μM; Fig. 4e). Hence, GM130 bears an IBB-like NLS sequence that structurally and functionally resembles an IBB domain, which allows it to separately interact with both full-length Importin-α and Importin-β.

**Phosphorylated GM130 and Ran reduce GM130-Importin−β binding**. A sub-stoichiometric ratio of Importin-α and Importin-β has been associated with membrane fractions of *Xenopus* egg extracts, has been pulled down by GM130 from HeLa cell lysates, and was recapitulated by our in vitro pulldown analysis (Supplementary Fig. 3e, g)[13,18]. These observations could result from: (1) phosphorylation on GM130 reducing its binding to Importin-β or (2) Ran-GTP in the extracts/lysates releasing GM130 from Importin-β but not from Importin-α. To test these hypotheses, we first carried out ITC to determine the $K_d$ value between GM130-NLS phospho-S25 peptide and Importin-β. Interestingly, binding of the GM130 phospho-peptide with Importin-β not only displayed an endothermic curve but also a binding constant that is at least three orders of magnitude lower than that of the wild-type GM130 peptide (Fig. 4f). This result suggests that the phosphor moiety on GM130 substantially reduces the binding affinity for Importin-β (see the "Discussion" section). Furthermore, our competition assays showed that the longer construct, GM130-S25D, was able to pull down Importin-β, and it could be displaced by IBB in this interaction (Fig. 4g). Hence, additional residues extending both N- and C-termini beyond the phosphopeptide (a. a. 13–40) may also be involved in Importin-β interaction and, importantly, the binding site on Importin-β for GM130 and IBB is mutually exclusive.

Next, we performed a GST pulldown experiment to examine whether GST-GM130 can interact with Importin-α or Importin-β in the presence of RanQ69L (a constitutively active mutant). We found that RanQ69L mediates release of Importin-β from Importin-β•GM130 complex and this is enhanced by the S25D

mutation (~70% in GM130-S25D vs. ~47% in GM130-WT; Fig. 4h, i). This result suggests that the phosphorylated GM130•Importin-β complex is more sensitive to RanQ69L than the wild-type complex. Moreover, Importin-β pulldown by GST-GM130-S25D was significantly reduced (~9-fold) in the presence of RanQ69L compared to when Ran was absent (Fig. 4j, k). However, under the same experimental conditions, GST-GM130-S25D pulled down comparable amounts of Importin-α FL with or without RanQ69L (Fig. 4j, k). Thus, the sub-stoichiometric pulldown ratios of Importin-α and Importin-β by GM130 (Supplementary Fig. 3e, g)[13,18] could arise because: (1) binding of GM130 to Importin-α FL is resistant to IBB displacement and is enhanced by GM130 phosphorylation; or (2) both the Ran pathway and GM130 phosphorylation suppress the interaction of GM130 and Importin-β.

**Importin-α-GM130 binding modulates mitotic Golgi disassembly**. Next, we introduced single point mutations at either the major (K34A) or minor (K14A) binding sites of GM130-S25D to establish whether they could disrupt binding of Importin-α FL. Importin-α FL was virtually undetectable when we used either of these GST-GM130 mutants to pull down recombinant protein or from HEK293T cell lysates (Fig. 5a–c, Supplementary Fig. 4a). Furthermore, binding assays revealed that GM130-S25D was able to bypass IBB auto-inhibition, with ~50% of Importin-α FL being pulled down by GM130-S25D relative to Importin-α(ΔIBB), which was substantially more than pulled down by the control (GST-IBB) (Supplementary Fig. 4b, c). Importantly, the ability of the K34A mutant to overcome IBB displacement was greatly reduced, i.e., to an extent even lower than for the control (Supplementary Fig. 4b, c). We also found that the K34A mutant could still pull down p115-CTD and Importin-β (Supplementary Fig. 4d), so the K34A single point mutation can disrupt the GM130 and Importin-α FL interaction without perturbing other known interactions.

To determine the cellular relevance of our biochemical findings, we used CRISPR/Cas9 genome editing to introduce the K34A point mutation of GM130 into a HEK293T cell line. Although no colonies representing homozygous mutations were found, we independently identified three cell lines with heterozygous mutations among ~500 screened colonies (Supplementary Fig. 4e). These three mutant colonies displayed a much slower growth rate and a higher percentage of G2/M cells (~5%) compared to the wild-type cell line (Fig. 5d–f), suggesting that the K34A mutation impedes mitosis and thereby reduces cell growth. To exclude that the phenotypes in the K34A mutant line were caused by reduced protein stability, we treated cells expressing wild-type or K34A-mutant GM130 with the protein synthesis inhibitor cycloheximide and measured their protein stabilities. Both wild-type and K34A mutant proteins were equally stable in cells, as indicated by the comparable band intensities in Western blots between these two proteins over a time-course (Supplementary Fig. 4f).

Since Golgi disassembly is required for mitotic progression[22], we next examined whether the K34A mutation can impair the Golgi disassembly process during mitosis. Wild-type and K34A mutant cells were stained with GM130 antibody and subjected to immunofluorescence (IF) analysis. Interestingly, the mutant cell lines presented a scattered, punctate pattern of greater fluorescence intensity during metaphase compared to the more evenly dispersed GM130 signal in wild-type metaphase cells (Fig. 6a, d, Supplementary Fig. 5a), implying that the K34A mutation does impair Golgi vesiculation. Two additional Golgi markers (GRASP65 and Giantin) were used to further validate this phenotype. Under the same IF experimental conditions, K34A

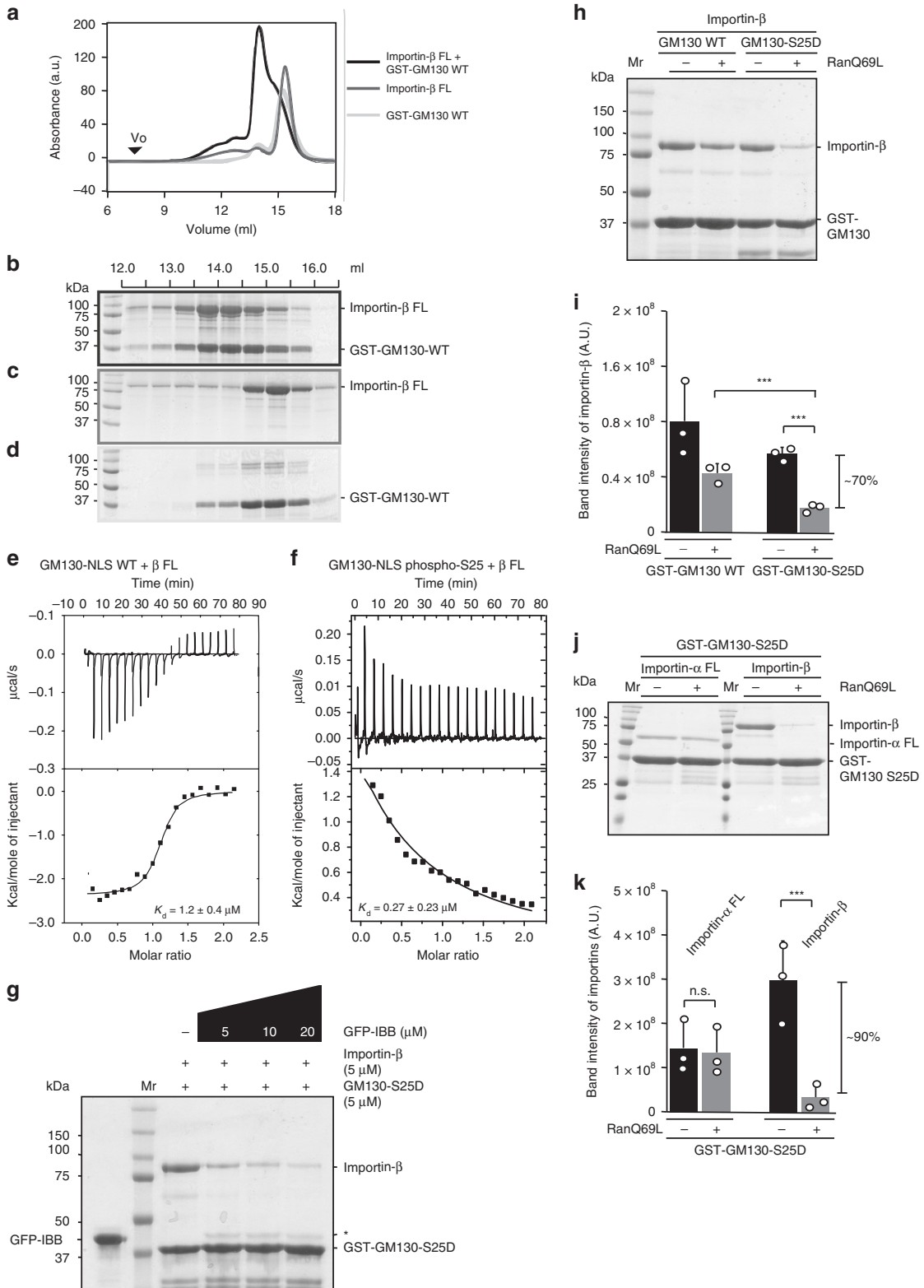

mutant metaphase cells stained by GRASP65, and Giantin antibodies displayed comparably elevated fluorescence intensities and punctate fluorescence patterns (Fig. 6b, c, e, f). We then arrested cells in prometaphase using nocodazole and examined p115 and GM130 protein–protein interactions using antibody-based PLA. During interphase, we observed punctate fluorescence concentrated near the nuclei of both wild-type and mutant cell lines, indicating that the vesicle–Golgi fusion facilitated by p115

and GM130 interaction is comparable for the wild-type and mutant protein (Fig. 6g, interphase cells). Only a few fluorescent dots could be observed in the wild-type mitotic cells (Fig. 6g, h), indicative of reduced vesicle–Golgi fusion. However, mitotic K34A mutant cells exhibited higher numbers of fluorescent dots compared to wild-type (Fig. 6g, mitotic cells), as evidenced by quantitative analysis (Fig. 6h). These results suggest that the p115 and GM130 interaction is enhanced in the K34A mutant cell line

**Fig. 4** Ran and a phosphate moiety repress GM130 and Importin-β interaction. **a** SEC (Superdex 200) elution profiles for the Importin-β•GST-GM130-WT complex (black line), Importin-β (gray line), and GST-GM130-WT (light gray line). The void volume (Vo) is indicated. **b–d** The peak fractions of the Importin-β•GST-GM130-WT complex (**b**), Importin-β (**c**), and GST-GM130-WT (**d**) were analyzed by SDS–PAGE and stained with Coomassie blue. **e, f** ITC titration curves (upper) and binding isotherms (lower) of Importin-β with GM130-NLS WT (**e**) and GM130-NLS phospho-S25 (**f**). $K_d$ values are indicated. **g** Competition binding assay. GST-GM130-S25D and Importin-β were incubated with indicated concentrations of GFP-IBB. Samples pulled down by GST beads were analyzed by SDS–PAGE and stained with Coomassie blue. The asterisk indicates the contamination of GFP-IBB in the pulldown. **h** GST pulldown assays of GM130-WT or GM130-S25D with Importin-β in the presence or absence of RanQ69L. **i** Analysis of the dissociation of Importin-β from either GM130-WT or GM130-S25D in the presence of RanQ69L. **j** GST pulldown assays of GM130-S25D with Importin-α FL or Importin-β in the presence or absence of RanQ69L. **k** Analysis of the dissociation of Importin-α FL and Importin-β from GM130-S25D in the presence of RanQ69L. For both GST pulldown assays, GST-fused GM130 (WT and S25D) was incubated with His-tagged RanQ69L and recombinant Importin-α FL or Importin-β at a molar ratio of 1:4:1 GM130:RanQ69L:Importin. GST-bound samples were analyzed by SDS–PAGE and stained with Coomassie blue. **i, k** Band intensities of Importin-α or Importin-β from the SDS–PAGE gels in **h**, **j** were used to determine averages of bound protein. Data represent mean ± standard deviation from three independent experiments. Differences were assessed statistically by two-tailed Student's t-test; ***p < 0.005. n.s. not significant

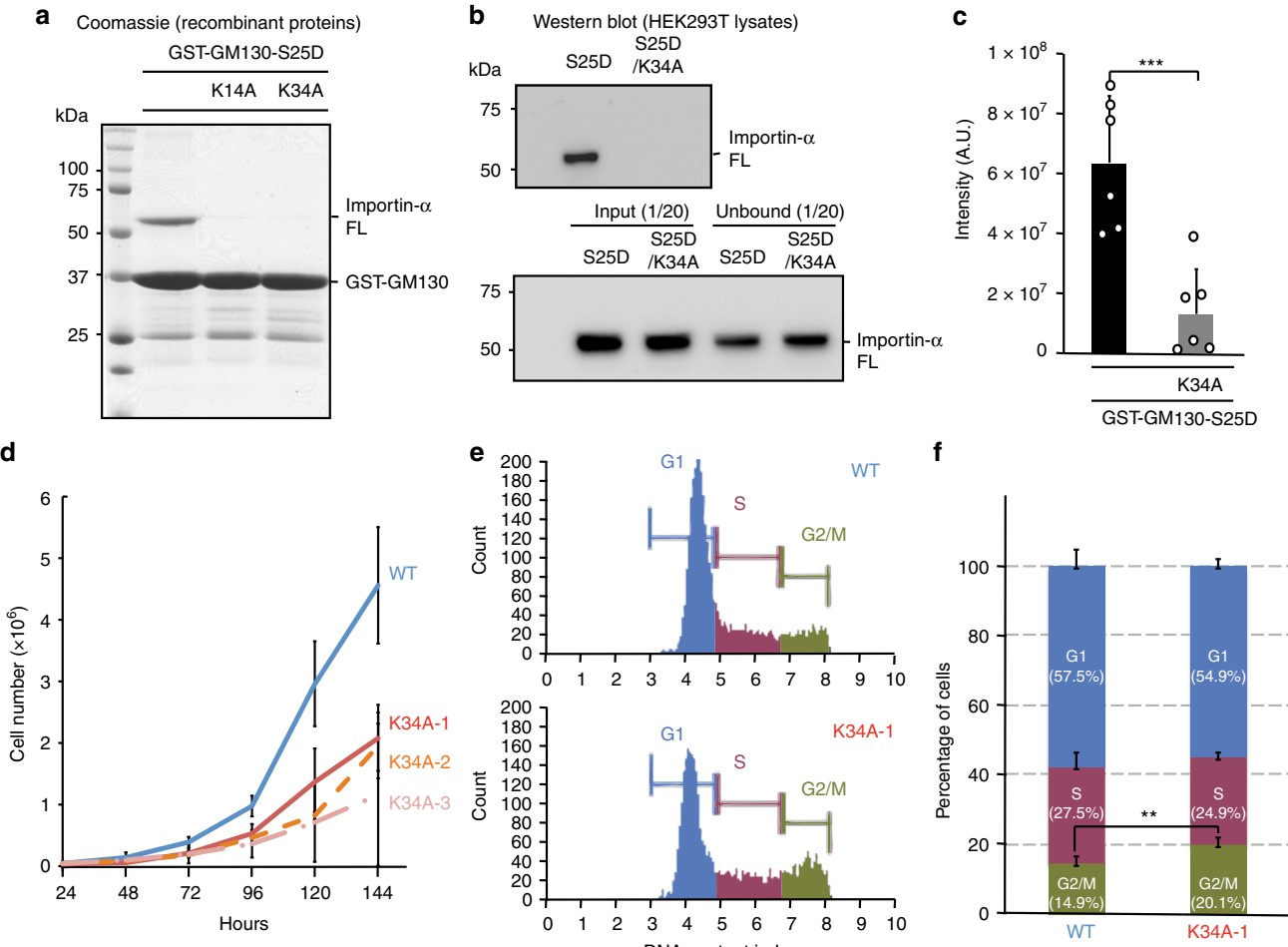

**Fig. 5** GM130 K34A mutation causes proliferative defects and G2/M arrest. **a** GST pulldown assays of Importin-α FL with GM130-S25D mutants. GST-fused GM130-S25D mutants were incubated with recombinant Importin-α FL. GST-bound samples were analyzed by SDS–PAGE and stained with Coomassie blue. **b** GST-fused GM130-S25D and GM130-S25D/K34A were incubated with HEK293T cell lysates. GST-bound samples were analyzed by SDS–PAGE followed by Western blotting using an Importin-α antibody. 1/20 of whole cell lysate (input) and of the total unbound proteins (unbound) are indicated. **c** Analysis of Importin-α FL pulled down by GM130-S25D from cell lysates. Band intensities of Importin-α FL from Western blotting in **b** were used to determine the averages of bound protein. Data represent mean ± standard deviation from more than three independent experiments. Differences were assessed statistically by two-tailed Student's t-test; ***p < 0.005. **d** Growth curves for wild-type (blue) and three K34A heterozygous mutant HEK293T (red, orange, and pink) cell lines. Equal numbers of cells were plated, and cells were counted each day for 6 consecutive days after plating. Data represent mean ± standard deviation from three independent experiments for each cell line. **e, f** Wild-type and K34A mutant cell lines were subjected to DNA content analysis. **e** DNA content of the wild-type (upper) and a K34A mutant cell line (lower) was measured using flow cytometry. Numbers of cells in the G1, S, and G2/M phases are shown in blue, purple, and dark green, respectively. **f** Percentages indicate cell cycle distributions of wild-type and mutant cells. Percentages of cells in the G1, S, and G2/M phases are represented by blue, purple, and dark green bars, respectively. Data represent mean ± standard deviation from three independent experiments. Differences were assessed statistically by two-tailed Student's t-test; **p < 0.01 for the G2/M phase in the wild-type and mutant cell line

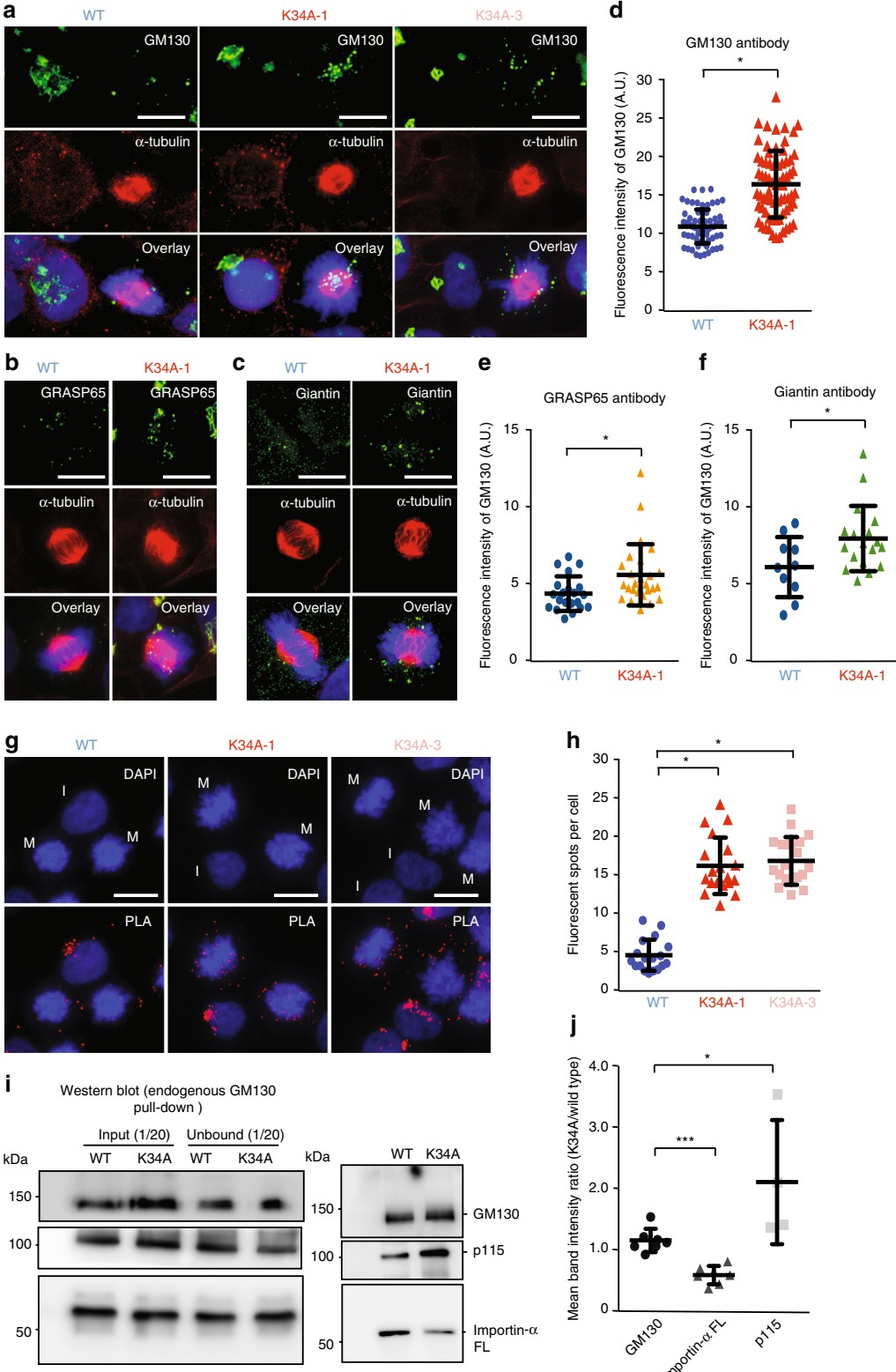

during the early stage of mitosis, so perturbation of mitotic Golgi fragmentation could lead to mitotic cell arrest.

Since wild-type and mutant GM130 presented equivalent protein amounts and comparable stability in cells (Supplementary Fig. 4f), we performed immunoprecipitation (IP) analysis to examine the amounts of p115 and Importin-α pulled down by endogenous GM130 from wild-type and K34A mutant cell lysates undergoing mitosis. Interestingly, we found that whereas

Importin-α pulldown by GM130 was lower in the mutant cell lysates relative to wild-type lysates, the mutant lysates exhibited elevated association of p115 with GM130 (Fig. 6i, j). These results suggest that the K34A mutation reduces binding of Importin-α by GM130, thereby allowing greater interaction between p115 and GM130, which facilitates p115-mediated vesicle fusion during the early phases of mitosis (prophase and metaphase). Furthermore, GM130 Ser-25 phosphorylation is carried out by cyclin-

**Fig. 6** Binding of Importin-α to GM130 regulates Golgi disassembly. **a–c** Representative immunofluorescence staining images of wild-type and K34A mutant cell lines using GM130 (**a**), GRASP65 (**b**), Giantin (**c**) (green, upper panel), and tubulin (red, middle panel) antibodies. The merged fluorescent images with DAPI staining are shown in the bottom panel. Scale bar: 10 μm. **d–f** Quantification plots of fluorescence intensities for GM130 (**d**), GRASP65 (**e**), and Giantin (**f**) in wild-type (blue) and K34A (red, GM130; orange, GRASP65; green, Giantin) mutant cell lines at metaphase. Mean and standard deviation were determined from data pooled from the three independent experiments. (More than 60 (GM130) and 20 (GRASP65 and Giantin) cells in total for each cell line.) Differences were assessed statistically by two-tailed Student's $t$-test; $*p < 0.05$. **g** Detection of endogenous p115 and GM130 interaction by PLA in wild-type (left panel) and two K34A mutant (middle and right panels) cell lines using p115 and GM130 antibodies. Red fluorescent dots are indicative of protein–protein interactions. Nuclei (blue) are shown by DAPI staining. Interphase (I) and mitotic (M) cells are indicated. Scale bar: 10 μm. **h** Quantification of numbers of fluorescent dots per cell in wild-type (blue) and mutant (red and pink) cell lines. Mean and standard deviation were determined from data pooled from three independent experiments (more than 45 cells in total for each cell line). Differences were assessed statistically by two-tailed Student's $t$-test; $*p < 0.05$. **i** Endogenous Importin-α and p115 were pulled down from nocodazole-arrested WT and K34A mutant cell lines using GM130 antibody. Pulldown samples were analyzed by Western blot using antibodies for GM130, Importin-α, and p115. 1/20 of whole cell lysate (input) and of the total unbound proteins (unbound) is indicated. **j** Analysis of Importin-α and p115 pulldown by GM130 from wild-type and mutant cell lysates. The scatter plot shows the ratios of GM130 (black), Importin-α (dark gray), and p115 (light gray) band intensities for K34A mutant cell lines over wild-type. Each dot represents an individual data point. Data represent mean ± standard deviation from at least four independent experiments (GM130 and Importin-α, $N = 7$; p115, $N = 4$). Differences were assessed statistically by two-tailed Student's $t$-test; $***p < 0.005$, $*p < 0.05$

dependent kinase 1 (Cdk1)[6,9], so we examined whether inhibition of Cdk1 activity could also alter the binding of Importin-α and p115 to GM130. Nocodazole-arrested cells were treated with the Cdk1 inhibitor RO-3306[23], followed by IP analysis of RO-3306-treated or untreated cell lysates using GM130 antibody. Inhibition of Cdk1 was assessed by changes in cell shape (Supplementary Fig. 5b), as 1-h RO-3306 treatment was sufficient to block Cdk1 activity, resulting in a morphological transition from rounded mitotic cells to a flat interphase shape[23]. Similar to what we observed in the K34A mutant (Fig. 6i, j), we also found that amounts of Importin-α and p115 pulldown by GM130 were reduced or enhanced, respectively, in mutant cell lysates (Supplementary Fig. 5c, d). Hence, we have revealed that fragmentation of the Golgi apparatus into vesicles, which is crucial for mitotic progress, is modulated by Importin-α-GM130 interaction, which is facilitated by Cdk1-mediated G130 phosphorylation.

**Importin-α-GM130 binding facilitates bipolar spindle assembly**. Next, we investigated whether impaired Golgi disassembly resulting from reduced Importin-α and GM130 interaction could cause mitotic spindle defects. The K34A mutant cell lines displayed a higher percentage of mitotic cells compared to that of wild-type (Fig. 7a), mirroring the higher proportion of G2/M cells in the mutant cell lines upon quantitation of DNA content (Fig. 5e, f). Moreover, the mutant cells showed a 3- to 4-fold increase in the percentage of mitotic cells with monopolar spindles relative to wild-type (Fig. 7d, e), suggesting that the K34A mutation on GM130 promotes monopolar spindle formation during mitosis.

Furthermore, we examined whether overexpression of the K34A mutant by means of the CMV promoter and depletion of endogenous GM130 by small interfering RNA (siRNA) can also increase the number of cells containing monopolar spindles during mitosis. Overexpression and knockdown efficiencies of GM130 were assessed by Western blot analyses (Supplementary Fig. 6a, b). Additionally, we also carried out IF staining to monitor whether Golgi-bound GM130 is depleted, as siRNA oligonucleotides partially knocked down GM130 (Supplementary Fig. 6b). Similar to observations reported by Kodani and Sutterlin[24], GM130 depletion resulted in mislocalization to Golgi of GRASP65 but not Giantin (Supplementary Fig. 6c–e), suggesting that Golgi-bound GM130 is substantially depleted.

Both GM130 mutant overexpression and endogenous GM130 depletion also resulted in elevation of mitotic cell percentages relative to controls, as found for the K34A mutant (Fig. 7a–c). Next, we assessed all mitotic cells to establish how many had

monopolar or multiple polar spindles (Fig. 7d). When GM130 was overexpressed (K34A mutant) or depleted, there was an equivalent ratio of monopolar and multiple spindle cells in the mitotic cells (Fig. 7g), even though the proportion of cells having monopolar spindles was slightly increased in the two concentrations of siRNA that mediated GM130 depletion (50 and 100 nM, Fig. 7f). These results suggest that knockdown of GM130 concurrently increases the numbers of mitotic cells containing monopolar and multiple polar spindles. Notably, in the K34A mutant cell lines, the mitotic cells with monopolar spindles were ~2- to 3-fold more abundant than cells with multiple polar spindles (Fig. 7g). Therefore, our findings imply that the monopolar spindle defect most likely is specifically enhanced by the point mutation on GM130. Notably, chemically inhibited Golgi disassembly has been shown to arrest cells with monopolar spindles[22]. Hence, we conclude that Importin-α−GM130-facilitated mitotic Golgi disassembly modulates spindle assembly and mitotic progression. Additionally, as GM130 has multiple important cellular functions in interphase and mitosis, other pathways through indirect or combined effects may cause the mitotic defect (i.e. the high percentage of mitotic cells) resulting from GM130 overexpression and depletion.

Finally, K34A mutant cells also presented more compact Golgi compared to the rather dispersed Golgi of wild-type cells in interphase upon staining with GM130 antibody (Supplementary Fig. 6f), suggesting that the Importin-α and GM130 interaction not only modulates Golgi disassembly but also Golgi reassembly during mitosis. The interplay of Importin-α, p115, and GM130 thus controls Golgi morphology during the cell cycle.

## Discussion

The Golgi apparatus undergoes fragmentation during mitosis, with p115-mediated Golgi fusion of transport vesicles being substantially reduced. Although GM130 phosphorylation has been correlated with dissociation of p115 from Golgi[5–7], the underlying molecular mechanism had not been characterized. GM130, a non-nuclear Golgi peripheral membrane protein, has been shown to bear an NLS-like sequence, allowing it to interact with Importin-α[13]. Importin-α negatively regulates many cellular activities during interphase and mitosis[25–27]. The mutually exclusive binding of p115 and Importin-α to GM130 (Figs. 1c, 3d–f) led us to hypothesize that Importin-α may suppress the p115 and GM130 interaction during mitosis, thereby impeding Golgi and vesicle fusion (Fig. 8a).

Conventionally, GTP-bound Ran binds to Importin-β to either directly dissociate cargos or by competing with the IBB domain (which self-binds to the major NLS-binding site of Importin-α) to

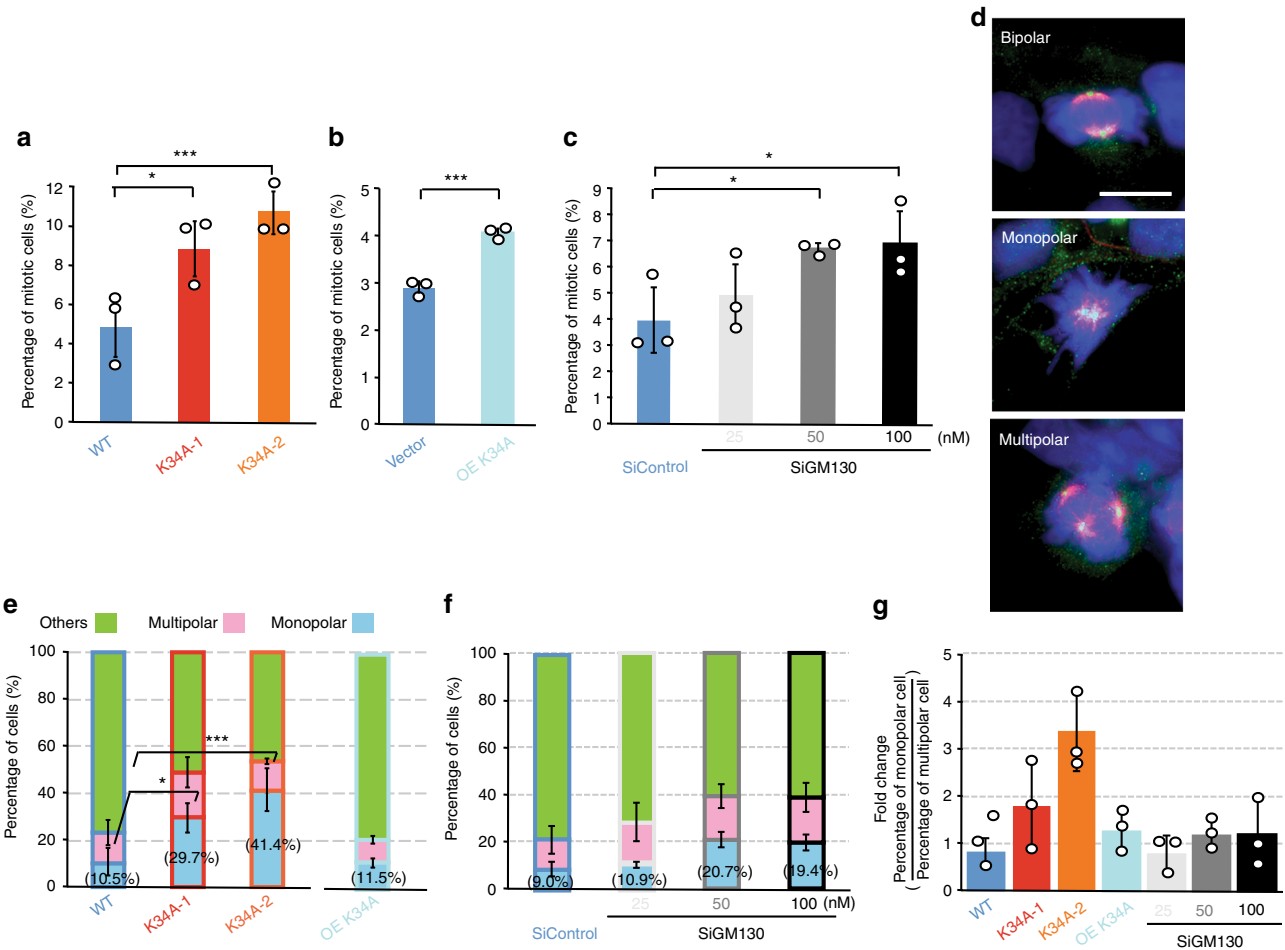

**Fig. 7** Perturbation of Golgi disassembly leads to monopolar spindle formation. **a–c** The indicated cell lines [K34A mutant cell lines (**a**), overexpression GM130 K34A mutant cell lines (**b**), and siRNA-mediated GM130 depletion cell lines (**c**)] were grown, fixed, and stained with α-tubulin antibody and DAPI. The percentages of mitotic cells in total cells of the control/WT (blue), mutant (red and orange), GM130 K34A mutant overexpressing (light blue), and GM130 siRNA treated (gray) cell lines are plotted. One-tailed Student's t-test; ***$p < 0.005$, *$p < 0.05$. Overexpression of the GM130 K34A mutant is indicated as OE K34A. Concentrations of siRNA oligonucleotides are indicated. **d** Representative merged fluorescent images of K34A mutant cells containing bipolar spindles (top), monopolar spindles (middle), and multiple polar spindles (bottom) at metaphase. Cells were immuno-stained with γ-tubulin antibody, α-tubulin antibody, and DAPI. Scale bar: 10 μm. **e**, **f** The percentages of cells containing monopolar and multiple spindles in total mitotic cells of the indicated cell lines are plotted. Percentages of monopolar spindle cells, multiple spindle cells and other mitotic cells are shown in light blue, pink, and light green bars, respectively. One-tailed Student's t-test; ***$p < 0.005$, *$p < 0.05$. **g** Ratios of monopolar spindle cells and multiple spindle cells of the indicated cell lines are plotted. Mean ± standard deviation of the percentages of the cells were calculated from $n = 3$ independent experiments. WT ($N = 1316$ cells); K34A-1 ($N = 963$ cells); K34A-2 ($N = 1064$ cells); Vector ($N = 4002$ cells); OE K34A ($N = 2539$ cells); SiControl ($N = 1488$ cells); SiGM130-25 ($N = 1193$ cells); SiGM130-50 ($N = 1331$ cells); SiGM130-100 ($N = 1021$ cells)

indirectly trigger the release of cargos. Although GM130 carries an IBB-like NLS that interacts with both full-length Importin-α and Importin-β, GM130 and Importin-β binding is perturbed by Ran and GM130 phosphorylation. Notably, the interaction between GM130 and Importin-α, which is enhanced by phosphorylation, is independently regulated by the Ran-Importin-β pathway, explaining why Importin-α but not Importin-β associates with GM130 in membrane fractions from either *Xenopus* egg extracts or mitotic mammalian cells[13,18]. More importantly, binding of Importin-α to phosphorylated GM130 can be sustained during mitosis in the presence of Ran-GTP, thereby ensuring that the kinase-phosphatase pathway instead of the Ran pathway controls the Importin-α interaction with GM130 during mitosis. Additionally, since GM130 functions as a membrane-associated higher-order structural assembly (e.g. tetramers[28]), the heterozygous K34A mutant colonies we isolated allowed us to study the role of this interaction beyond microtubule

nucleation[13]. The heterozygous GM130 mutant may retain the ability to partially compete for Importin-α due to high local protein concentration, thereby minimizing defects in the microtubule nucleation function of GM130[13]. However, GM130 complex containing either wild-type or mutant peptides can interact with p115, still enabling p115-mediated vesicle fusion during mitosis and leading to impairment of Golgi disassembly and a higher percentage of monopolar spindles.

Our crystal structure of the Importin-α•GM130-NLS complex showed that three regions of the GM130-NLS are involved in binding Importin-α (Fig. 1i). The positively charged residues of GM130-NLS facilitate interactions with the major (five lysines) and minor (three lysines) NLS-binding sites of Importin-α and, as a result, GM130 has a larger interaction network compared to that of the classical NLS (Supplementary Fig. 1a; IBB and TPX2). Interestingly, apart from the two clusters of basic residues that bind to the major and minor NLS-binding sites, the intervening

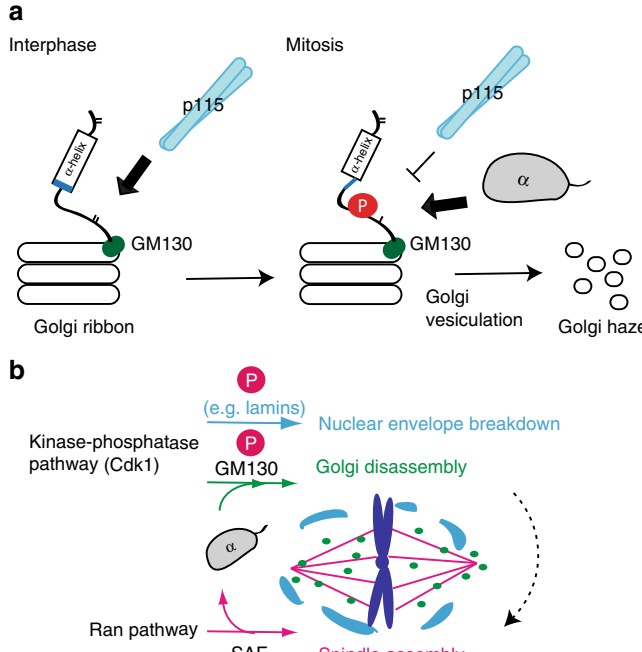

**Fig. 8** A schematic model for how Importin-α regulates mitotic Golgi disassembly. **a** During interphase, p115 facilitates vesicle fusion to the Golgi ribbon by binding to GM130. During mitosis, binding of Importin-α to GM130 is enhanced by mitotic phosphorylation, leading to steric hindrance of the p115–GM130 interaction and thereby suppressing p115-mediated vesicle fusion, which allows Golgi to undergo vesiculation. **b** Schematic for how Importin-α coordinates the Ran and kinase-phosphatase pathways to regulate major events of mitosis (NEBD, Golgi disassembly, and spindle assembly). Cdk1 phosphorylates numerous proteins in the nuclear envelope and GM130 in Golgi. Phosphorylation of GM130 enhances binding of the Importin-α that is released from SAFs by Ran, prohibiting p115 binding to GM130, and promoting Golgi disassembly. Golgi disassembly then ensures proper spindle assembly (black dashed line with an arrow). Spindle assembly factor (SAF) is indicated

linker region of GM130 contains an α-helix that also interacts with Importin-α, resulting in a much more extensive overall interaction interface between GM130-NLS and Importin-α compared to other interacting partners (e.g. IBB and TPX2) (Supplementary Fig. 1a). Thus, GM130-NLS exhibits a non-classical interaction mode with Importin-α. Moreover, the phosphate moiety on GM130 does not directly enhance the binding affinity between GM130 and Importin-α. Instead, our crystal structures of Importin-α in complex with either a phosphomimetic or phosphorylated GM130 peptide revealed increasing numbers of contacts with the minor NLS-binding site of Importin-α and additional intramolecular interactions within GM130 (Supplementary Fig. 1a).

MD simulations further revealed destabilization of the free GM130 peptide in solution upon phosphorylation, in addition to water-mediated intramolecular interactions of the bound phosphorylated peptide. Most cNLS (e.g. IBB) sequences lack secondary structural elements and are likely to be unstructured within proteins[29]. However, the GM130 NLS contains an α-helix that may interfere with the GM130 and Importin-α interaction. Hence, phosphorylation of apo-GM130 shortens the α-helix, promoting GM130 and Importin-α binding (Fig. 2g). Notably, IBB uses an unfolded configuration to interact with Importin-α, but adopts a folded α-helix conformation in complex with Importin-β[29]. Hence, destabilization of the apo-GM130 peptide α-helix upon phosphorylation may result in the reduction of

binding affinity between GM130 and Importin-β (Fig. 4e, f). Previous studies have reported that phosphorylation may either destabilize or stabilize α-helix in the free peptides, therefore modulating biological activities of proteins[30,31]. Hence, we propose that phosphorylation plays a regulatory role in two ways: (1) by destabilizing the conformation of free GM130-NLS and (2) by stabilizing the conformation of GM130-NLS bound to Importin-α.

The NE undergoes breakdown (NEBD) at the onset of open mitosis, requiring phosphorylation of numerous proteins residing in the NE (e.g. nuclear lamins) by Cdk1[32]. Notably, Cdk1 also phosphorylates the Ser-25 residue within the NLS-like motif of GM130 during early mitosis, and this GM130 phosphorylation event has been correlated with mitotic Golgi disassembly[6,9]. Here, we have demonstrated that phosphorylation of Ser-25 enhances the GM130 and Importin-α interaction, thereby sterically blocking the p115-binding site of GM130. During early mitosis, RanGTP releases Importin-α/-β from spindle assembly factors (SAFs) (e.g. TPX2[27] and NuMA[26]), activating the mitotic functions of these SAFs and in turn promoting mitotic spindle assembly. Thus, the Importin-α liberated by RanGTP can instead inhibit the p115-binding function of Cdk1-phosphorylated GM130, suppressing p115-mediated vesicle–Golgi fusion and allowing Golgi disassembly. Taken together, our findings provide a model whereby the Ran and kinase-phosphatase pathways coordinately regulate major events during mitosis (NEBD, Golgi disassembly, and spindle assembly) to ensure mitotic progression, facilitated by Cdk1 and the nuclear transport factor Importin-α (Fig. 8b).

## Methods

**Bacterial expression constructs.** *Homo sapiens* GM130 (a.a. 1–48), GM130 (a.a. 1–85), and Importin-β were amplified by PCR and ligated into pGEX-6P-1 (GE Healthcare). *H. sapiens* full-length Importin-α and *Mus musculus* Importin-α were amplified by PCR and ligated into modified pGEX-6P-1 (GE Healthcare) that produces proteins with 6-His at the C–terminus. Mouse Importin-α shares ~95% identity with human Importin-α over the full protein sequence, so we used the former for biochemical and structural studies. GM130-S25D (a.a. 1–48), GM130-S25D (a.a. 1–85), GM130-S25D/K14A (a.a. 1–85), and GM130-S25D/K34A (a.a. 1–85) mutants were created by PCR-based mutagenesis and were verified by sequencing. A list of primer sequences used in this study is shown in Supplementary Table 1.

**Protein expression and purification.** Bacterial expression constructs were transformed into *Escherichia coli* Rosetta strain (Novagen). Protein production was induced by 0.5 mM IPTG at 18 °C overnight. The cell pellet was re-suspended, disrupted by French press and clarified by centrifugation at $15,000 \times g$ at 4 °C for 20 min. Recombinant proteins were then purified by chromatography.

Cells containing GST-tagged GM130 (a.a. 1–48)•His-Importin-α (a.a. 70–498) were harvested and re-suspended in a buffer containing 20 mM HEPES (pH 7.4), 150 mM NaCl, and 3 mM DTT. Clarified lysate was loaded onto a GST column (GE Healthcare) and protein samples were eluted by a buffer containing 20 mM HEPES (pH 7.4), 150 mM NaCl, 50 mM reduced glutathione, and 3 mM DTT. The GST-GM130•Importin-α(ΔIBB) complex was incubated with PreScission protease at 4 °C overnight to remove the GST tag. PreScission-digested samples were incubated with nickel resin (Sigma), followed by a pre-wash of 25 mM imidazole. Next, protein was eluted by a buffer containing 250 mM imidazole and then subjected to SEC (Superdex 200 16/600) analysis. The peak fractions were analyzed by SDS–PAGE. Purified protein fractions were collected and concentrated to 25 mg/ml using Amicon Ultra 15 (Millipore) for crystallization. Purification of the GM130-S25D (a.a. 1–48)•Importin-α (a.a. 70–498) complex was carried out using the same method. To produce the GM130-NLS phospho-S25•His-Importin-α (ΔIBB) (a.a. 70–498) complex for crystallization, GST-tagged His-Importin-α (a.a. 70–498) was purified by GST-, nickel- and SEC chromatography using the same protocol for the purification of other GM130-containing complexes. The Importin-α protein was then mixed with a synthetic GM130-NLS phospho-S25 peptide (synthesized by the peptide synthesis core facility of Academic Sinica) in a molar ratio of 1:3. The excess peptide was removed by loading the sample into SEC (Superdex 200 10/300). Purified protein fractions were collected and concentrated to 25 mg/ml using Amicon Ultra 15 (Millipore) for crystallization. The same expression and purification protocol was applied to purify GST-tagged GM130-WT, GM130-S25D, GM130-S25D/K14A, and GM130-S25D/K34A, as well as non-

tagged Importin-β, GM130-WT, and GM130-S25D. GST tags were removed by PreScission protease cleavage.

To purify p115 (a.a. 780–930), cells containing p115 (a.a. 780–940) were resuspended in a buffer containing 50 mM K-phosphate (pH 7.4), 150 mM NaCl, and 2 mM β-mercaptoethanol. The cells were lysed with a cell disrupter. After clarification, lysates were applied to nickel resin (Sigma), followed by a pre-wash of 25 mM imidazole. Proteins were eluted with a buffer containing 250 mM imidazole. Fractions containing p115 protein were pooled and loaded into SEC (Superdex 75 16/600) and analyzed by SDS–PAGE. Purified protein fractions were collected and concentrated using Amicon Ultra 15 (Millipore). RanQ69L and GFP-Importin-α-NLS were obtained using an identical expression and purification protocol.

**Protein crystallization and structure determination**. The purified GM130 (a.a. 1–48)•Importin-α (a.a. 70–498) complex was screened for crystallization. Crystals were grown at a concentration of 25 mg/ml at 20 °C using the hanging drop method. Crystals grew to their maximum size after one week in 0.1 M sodium malonate and 12% w/v polyethylene glycol 3350 (pH 7.0). Crystals grew in the orthorhombic space group $P2_12_12_1$ and contained a heterodimer in the asymmetric unit. For cryoprotection, crystals were stabilized in 0.1 M sodium malonate, 12% w/v polyethylene glycol 3350 (pH 7.0), and 20% (v/v) glycerol.

Crystals of the GM130-S25D (a.a. 1–48)•Importin-α (a.a. 70–498) and GM130-NLS phospho-S25 (a.a. 13–38)•Importin-α (a.a. 70–498) complexes were also grown at concentrations of 25 mg/ml at 20 °C using the hanging drop method. Crystals of these two complexes grew in 0.2 M potassium thiocyante and 20% w/v polyethylene glycol 3350 (for GM130-S25D (a.a. 1–48)•Importin-α (a.a. 70–498)) or 0.1 M sodium citrate tribasic dihydrate and 16% w/v polyethylene glycol 8000 (pH 5.5) (for GM130-NLS phospho-S25 (a.a. 13–38)•Importin-α (a.a. 70–498)), respectively.

X-ray data were collected at beamline TPS 05A and BL13B1 at NSRRC (Taiwan, ROC) using a wavelength of 0.99984 Å. X-ray intensities were processed using HKL2000[33], and molecular replacement was performed using Phaser in the Phenix software, with Importin-α (PDB code: 1IQ1 [https://www.rcsb.org/structure/1iq1]) as a search model[34]. For all of these structures, electron densities for GM130 could only be observed and assigned for residues 13–38 or 39. No clear electron densities outside this region could be identified. All the residues were in favored or allowed regions, and thereby no residues in the disallowed region of the Ramachandran plot. Coordinates and structure factors have been deposited in the Protein Data Bank.

**GST pulldown analysis using purified recombinant protein**. A 1:1 molar ratio of GST-GM130-WT or mutant GM130 and Importin-α FL was incubated with glutathione-coupled sepharose beads (GE Healthcare). After pre-washing with 150 mM NaCl, proteins were eluted by a buffer containing 50 mM reduced glutathione and analyzed by SDS–PAGE. To examine the interactions between GST-GM130-S25D and Importin-α FL or Importin-β in the presence of RanQ69L, we incubated GST-GM130-S25D, RanQ69L and Importin-α FL or Importin-β in a molar ratio of 1:4:1.

**Size exclusion chromatography**. GST-GM130-WT and GST-GM130-S25D were, respectively, incubated with Importin-β or Importin-α FL in a molar ratio of 1:1. Samples were injected into an SEC (Superdex 200 16/60) column (GE Healthcare) pre-equilibrated with a buffer containing 20 mM HEPES (pH 7.4), 50 mM NaCl, and 3 mM DTT. Peak fractions were analyzed by SDS–PAGE.

**Isothermal titration calorimetry**. Binding affinities of different GM130 proteins to Importin-α and Importin-β were measured by ITC (MicroCal iTC200). All proteins were dialyzed against ITC buffer (20 mM HEPES pH 7.4, 1 mM DTT) with either 50 or 200 mM NaCl. Protein concentrations were determined spectrophotometrically with respective extinction coefficients. ITC was performed at 25 °C. The respective heat changes were fitted using a one-sites binding model.

**CD spectra thermal melting assays**. The CD spectra of the GM130-WT(1–48)•Importin-α (a.a. 70–498), GM130-S25D(1–48)•Importin-α (a.a. 70–498), and GM130-NLS phospho-S25 (a.a. 13–38)•Importin-α (a.a. 70–498) complexes were recorded using an AVIV CD400 spectrometer. Thermal melting temperatures were determined by measuring the ellipticity at a wavelength of 222 nm as a function of temperature from 20 to 75 °C. The protein concentration used in the CD measurements was 0.2 mg/ml in a buffer of 20 mM HEPES (pH 7.4), 50 mM NaCl, and 3 mM DTT. Measurements were taken three times and averaged.

**Molecular dynamics**. Starting from the crystallographic structure of the native GM130-NLS•Importin-α complex, eight 200-ns simulations at 300 K and 1 atm were performed for the apo peptides (native and phosphorylated GM130-NLS), whereas ten 50-ns simulations were performed for the GM130-NLS•Importin-α and GM130-NLS phospho-S25•Importin-α complexes. All histidines were found to be neutral according to Propka 3.1 and were thus deprotonated[35]. Each system was solvated in a box of TIP3P[36] water molecules at an ionic strength of 0.15 M using

Na[+] and Cl[−] ions. All bonds to hydrogen atoms were constrained by the SHAKE[37] algorithm. Long-range electrostatic forces were treated using the particle mesh Ewald method[38] with a grid spacing of 1 Å and a cutoff of 12 Å. The non-bonded interactions were updated every 1 fs. The simulations employed the CHARMM36 force-field[39] using the NAMD 2.12[40] program with an integration time step of 2 fs. For each complex simulation, the first 10 ns was excluded and hydrogen bonds were analyzed using the CHARMM37 program[41] and an acceptor to hydrogen distance of ≤2.4 Å and a donor–hydrogen–acceptor angle of ≥130°. Each 200-ns apo simulation was divided into 1 ns sub-windows from which 200 conformations were extracted and the secondary structure of each residue was assigned using DSSP[42].

**Growth curves**. HEK293T cells (ATCC®CRL11268) were cultured in Dulbecco's modified Eagle medium (DMEM) with 10% fetal bovine serum and 1% penicillin and streptomycin. Cells were seeded in a 24-well plate with a concentration of 5000 cells per well. Cell numbers were determined by counting trypan blue-negative cells using a hemocytometer under microscopy. Cells were counted every 24 h after seeding, and growth curves were generated based on three independent experiments.

**Cell-cycle analysis**. The cell-cycle analysis was measured using a Muse Cell Cycle Assay Kit (MCH100106, Merck, Millipore) according to the manufacturer's instructions. In brief, cells were dissociated and washed once with 1X PBS and fixed with 70% ethanol overnight. Cells were then centrifuged and washed once with 1X PBS. Cells were then re-suspended in 200 μl of Muse cell cycle assay kit and analyzed using Muse Cell Analyzer software (Merck, Millipore).

**Immuno-precipitation**. HEK293T cells were lysed in RIPA buffer containing 50 mM Tris, 150 mM NaCl, 1% NP-40, 1% sodium deoxycholate, 0.1% SDS, 1 mM phenylmethyl sulphonyl fluoride, 1 μg/ml aprotinin, and 1 μg/ml leupeptin. Supernatants were collected after centrifugation at 21,000×g for 10 min at 4 °C. Cell lysates were incubated with 5 μM of GST-GM130-WT, GST-GM130-S25D, or GST-GM130-S25D/K34A, and glutathione-coupled sepharose beads (GE Healthcare). After three pre-washes with buffer containing 150 mM NaCl, proteins were eluted using a buffer containing 50 mM of reduced glutathione. Eluted proteins were analyzed by SDS–PAGE, followed by Western blotting using antibodies against GM130 (Abcam ab52649), Importin-α (R&D Systems MAB6207), and Importin-β (Abcam ab2811). Uncropped scans of Western blots are provided in Supplementary Fig. 7.

To pull down endogenous GM130, HEK293T cells were synchronized via nocodazole treatment and lysed in a buffer containing 50 mM Tris–HCl (pH 7.4), 150 mM NaCl, 1 mM EDTA, 10% glycerol, 1% Triton, 0.15% SDS, protease inhibitor cocktail (Roche), and 1 mM PMSF. Clarified lysates were pre-incubated with GM130 antibody (Abcam ab52649) for 12 h at 4 °C, followed by one hour incubation with IgG sepharose™ 6 Fast Flow (GE Healthcare). Beads were then washed three times with buffer (20 mM HEPES pH 7.4, 150 mM NaCl, 1.5 mM MgCl$_2$, 10% glycerol, 0.1% Triton). Protein samples were boiled in SDS sample buffer and analyzed by SDS–PAGE, followed by Western blotting analysis.

To prepare Cdk1 inhibitor treated lysates for GM130 pull-down assays, WT HEK293T cells were synchronized by 30 μM of nocodazole for 13 h, followed by treatment of Cdk1 inhibitor RO-3306 (10 μM/ml) (Sigma, cat#SML0569) for 2 h.

**CRISPR/Cas9 targeting strategy**. SgRNA targeting human *Golga2* was designed using the Optimized CRISPR Design website (http://crispr.mit.edu) followed by ligation into PX330 vector (Addgene 42330). HEK293T cells were dissociated as single cells using 0.05% Trypsin-EDTA. We co-transfected $1 \times 10^6$ million cells with 4 μg Cas9-sgRNA plasmid, 400 nM single-stranded DNA and a puromycin vector (pLVX-mCherry-C1) using Nucleofector Kit 2 (Lonza VPH-5022). Cells were then selected with puromycin. Approximately 10 days later, we selected single colonies for expansion and subsequent genotype sequencing. Sequences of the oligonucleotides used in the CRISPR/Cas9 genome editing experiment are shown in Supplementary Table 1.

**IF staining**. We used GM130 (1:1000, rabbit; Abcam ab52649), GRASP65 (1:500, rabbit; Abcam ab174834), Giantin (1:500, rabbit; Abcam ab37266), α-Tubulin (1:1000, mouse; GeneTex GXT628802), and Importin-α (1:1000, mouse; R&D Systems MAB6207) antibodies, followed by Alexa Fluor 488-conjugated and Alexa Fluor 568-conjugated donkey secondary antibodies against mouse or rabbit (1:1000, Invitrogen). Nuclei were counterstained with DAPI. Fluorescence images were collected using an Olympus fluorescence microscope (IX83), 3D images were collected using a Zeiss confocal microscope (LSM780), and all images were analyzed using Imaris software.

**Proximity ligation assay**. Cells were synchronized with nocodazole and fixed with 4% PFA. Fixed cells were then stained with Duolink (cat# DUO92008, Sigma). Our PLA protocol was performed according to the manufacturer's instructions using GM130 (Abcam ab52649, 1:25) and p115 (Abcam H00008615-M03, 1:25)

antibodies. Nuclei were counterstained with DAPI. Images were taken using an Olympus fluorescence microscope (IX83). Data were quantified by counting fluorescent spots.

**Small interference RNA knockdown of GM130 (GOLGA2).** GM130 SMART pool and Non-Targeting siRNAs were purchased from Dharmacon. HEK293T cells were transfected with siRNA oligonucleotides using Lipofectamine RNAiMax (Invitrogen, cat#13778) according to the manufacturer's protocol. After 24-h transfection, transfected cells were replated on coverslides for further IF staining after 24-h seeding.

**Overexpression of GM130 K34A mutation.** Full length of GM130 K34A mutation was cloned into Lenti-virus vector pCDH-CMV-MCS-EF1-GFP-T2A-Puro (BioCat, Cat#CD513B-1). For virus production, HEK293T cells were plated at $1.5 \times 10^6$ per well into six-well plates. Next day cells were co-transfected with Lenti-viral GM130 K34A and lentiviral packaging plasmid psPAX2 (Addgene# 12260) and pMD2.G (Addgene #12259) using a standard $CaCl_2$ transfection method described previously[43]. After 48 h, virus supernatants were collected and passed through 0.45 μm pore size filters (Millipore, Billerica, MA, www.millipore.com). GM130 K34A mutation virus and 4 μg/ml of polybrene (Sigma) were added to the HEK293T cells. Infection efficiency was enhanced by spinning the plates at $1100 \times g$ for 30 min. Infected cells surviving puromycin selection (2 μg/ml) were then expanded and harvested for further analysis.

**Cycloheximide treatment.** GM130 WT and K34A mutant cells were replated on six-well plates. Next day cells were treated with 100 μg/ml of cycloheximide (Sigma, cat#C7698) for 0, 4, and 8 h.

**Reporting summary.** Further information on research design is available in the Nature Research Reporting Summary linked to this article.

## Data availability

The data supporting the findings of this manuscript are available from the corresponding author upon reasonable request. The source data underlying Figs. 1a, 1b, 1d–f, 2e, 3a, 3b, 3e, 3f, 4e, 4f, 4i, 4k, 5c, 5d, 5f, 6d–f, 6h, 6j, 7a–c, 7e–g and Supplementary Figs. 1d–f, 4a, 3f, 3g, 4c, and 5d are provided as a Source Data file [https://doi.org/10.6084/m9.figshare.6025748]. Coordinates and structure-factor files for GM130-WT, GM130-S25D, and GM130-pS25 have been deposited in the Protein Data Bank, with accession codes 6IW8 [https://www.rcsb.org/structure/6IW8], 6K06 [https://www.rcsb.org/structure/6K06], and 6IWA [https://www.rcsb.org/structure/6IWA], respectively.

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

## Acknowledgements
We appreciate the technical services provided by the "Synchrotron Radiation Protein Crystallography Facility of the National Core Facility Program for Biotechnology, Ministry of Science and Technology" and the "National Synchrotron Radiation Research Center", a national user facility supported by the Ministry of Science and Technology of Taiwan, ROC. We acknowledge the use of the ITC system in the Biophysics Core Facility, funded by Academia Sinica Core Facility and Innovative Instrument Project (AS-CFII108-111). K.-C.H. acknowledges support from the Ministry of Science and Technology (MOST-106-2311-B-001-038-MY3) and Academia Sinica (AS-CDA-106-L02). S.-Y.T. acknowledges support from the Ministry of Science and Technology (MOST-107-2320-B-002-026 and MOST-107-2314-B-002-079). C.L. acknowledges support from the Ministry of Science and Technology (MOST-107-2113-M-001-018) and Academia Sinica (AS-IA-107-L03).

## Author contributions
C.-C.C. performed biochemical experiments. C.-J.C., Y.-C.P. and S.-Y.T. carried out cell-based experiments. C.-C.C. and K.-C.H. determined crystal structures. C.G. and C.L. performed molecular dynamics experiments. All authors analyzed the data, discussed the results and helped write the manuscript. S.-Y.T. and K.-C.H. directed the project. K.-C.H. prepared the manuscript.

## Additional information

**Competing interests:** The authors declare no competing interests.

