## [Peer Review File · Nature Communications]

Reviewers' comments:

Reviewer #1 (Remarks to the Author):

This is an interesting paper that describes a potentially novel function of importin alpha as a regulator of Golgi disassembly. Unfortunately, the paper is unusually concise and falls short in describing important experimental details. I have some general comments that the authors should consider and address in order to make this work better readable and scientifically sound.

Major points

1. Title. It is not clear to me why this process would be Ran-independent. As long as GM130 binds to both importin alpha and importin beta, RanGTP is still involved. I don't see any experiment in the paper that rules out this possibility
2. The Introduction is skimpy and fails to reference important published work on membrane protein NLSs. In addition, it doesn't touch upon the broad field of nuclear transport.
3. Crystallography. The R_{work}/R_{free} values for GM130-WT and GM130-S25D are unacceptably high (the R_{free}) and uncoupled raising questions on the quality of the NLS model. I ask the authors to do three things:
First, the authors refine their model and lower the R_{free} by at least 5-7 points (at least);
Second, they should show an OMIT map for the NLS, especially the novel alpha helix feature;
Third, in the crystallographic table (that should be in the main text) they should provide the B-factor for the data (Wilson b-factor), importin alpha model, NLS, and water.
4. Phosphorylation at GM130-S25. It is not clear how the phosphate at position 25 of GM130 is stabilizing the NLS-helix. This should be shown in greater detail and substantiated by experimentally or computationally (e.g., Stability curves, Molecular Dynamics simulation, etc.)
5. GM130-NLS binding to importin beta. Does phosphorylation of S25 affect binding to importin beta? This should be shown in Figure 3d.
6. GM130-NLS competition with IBB. Carry out a competition assay between GM130-NLS and IBB-domain to determine if binding to importin b is mutually exclusive. Also, in reference to point #1, why would the process Ran-independent?
7. Affinity for FL-Importin alpha. The affinity of GM130-NLS for importin beta is 1.2 uM, and there's some ~3 uM importin b in a cell, which means GM130 will be mostly bound to importin beta at steady state. In contrast, the measured affinity of GM130-NLS for FL-importin alpha is 6.9uM (2.1uM after phosphorylation), but the estimated concentration of importin alpha is ~1uM. Thus, what is the biological significance of focusing this paper on a 'novel' property of importin alpha which is not likely to be significant in vivo?
8. Role of Ran. Section "Conventionally, GTP-bound Ran binds to importin b..." There seems to be a misunderstanding in the way the authors describe the putative role of Ran on importin a:Gp130-NLS interaction. The authors refer to this interaction as Ran-independent and even the title hints at this. However, NLS-binding to importin a is always Ran-Independent. The authors are confusing RanGTP-mediated displacement of importin alpha:NLS from importin alpha with the auto-displacement of the NLS from importin alpha that is mediated by the IBB. What the authors need to demonstrate here is that the GM130-NLS is immune from IBB-autoinhibition, and that is not likely to be the case given the K_d measured by ITC. The authors should set up a pull-down like in Fig 1c that has as control DIBB-importin alpha. The quantity of NLS-cargo pulled-down by FL-importin alpha has to be normalized to the total GM130-NLS captured by the DIBB-importin alpha.

9. Figure 3f. Repeat the RanQ69L displacement pull-down in the presence of physiological concentrations of importin beta (~3uM) and FL-importin alpha (~1uM).

Overall, this paper is not presentable in the current form but needs additional experiments and controls.

Reviewer #2 (Remarks to the Author):

The paper by Chih-Chia Chang et. reveals a novel mechanistical aspect of the regulation of mitotic Golgi disassembly. In particular, it has been known for more than two decades that Golgi disassembly is also induced by inhibition of membrane fusion in consequence of suppression of the interaction of the Golgi matrix proteins p115 and GM130. This inhibition correlates with mitotic phosphorylation of GM130. However, the mechanism through which phosphorylation affects the binding of the proteins, and the physiological consequences of this inhibition, were not known.

Here Chang and co-workers show the unexpected and interesting finding that inhibition p115/GM130 interaction is mediated by the binding of Importin- α (a nuclear transport factor) to GM130.

By using a series of biochemical approaches, the authors show that Importin- α directly competes with p115 for the binding to GM130. The binding of Importin- α is promoted by phosphorylation of Ser25 of GM130. By co-crystallization and modelling experiments, the authors identify the minimal and essential requirements for Importin- α /GM130 interaction. Based on this, they design GM130 point-mutants that are not able to bind Importin- α , and they show that cells expressing the mutant displayed large Golgi aggregates during mitosis and mitosis defects, in line with a less effective disruption of the interaction of p115 with GM130.

In general, I think that the paper is interesting and reveals a novel mechanism of regulation of Golgi fragmentation during mitosis, and they also suggest an intriguing possible coordination with nuclear disassembly. I think that the general conclusion of the paper are plausible and should be of interest to a wide audience. In general, the paper is technically sound, but it is missing a series of important controls and quantifications. Moreover, I suggest that a deeper investigation of the physiological consequences of alterations of the binding of importin to GM130 would strengthen the paper. Thus, I suggest that the following points should be addressed before publication.

Major concerns:

1) The conclusions based on the pull-down or co-IP experiments appear plausible and are in line with the other approaches. Yet, the experiments are missing important controls. The western blots of Fig 3c, 4h and extended 2e should also show the amount of the proteins of interest before ("input") and after ("unbound") the pull-down or IP. Moreover, the results should be expressed in a quantitative manner; this is particularly important when comparing the differential co-IP of importin- α and importin- β with GM130. In addition, as the in vitro interaction of importin- α with GM130 appears to be regulated by phosphorylation (most probably by Cdk1), it would be interesting to test the effect of Cdk1 inhibition on the interaction of the endogenous proteins.

2) The investigation of the physiological consequences of the GM130 mutation appears to be limited; one concern is related to the use of the few heterozygous colonies survived during the CRISPR/cas9 genome editing procedure. I think that a similar set of experiments should be performed in cells depleted by siRNA and/or expressing the GM130 mutants that do not bind to importin- α ; as a readout, the authors should examine Golgi fragmentation (in a quantitative manner) and mitotic defects. In addition, I think that it is important to try to address the core mechanism for the mitotic defect. For example, one possible cause is the reduced level of Golgi fragmentation; the other could be that the GM130 mutant remains bound to p115 and, as a consequence, does not work properly in stabilizing the mitotic spindle (Wei et al J Cell. 2015). Have the authors tested if the cells expressing GM130 mutant show spindle defects?. I think that the authors should elaborate more, in terms of experiment

and discussion, on this aspect.

3) Elaborate and discuss more on the model. For example, is the binding of Importin- α to GM130 correlated with nuclear envelope breakdown?. This could suggest a functional coordination between nuclear envelope breakdown and disassembly of the Golgi stacks and formation of the spindle.

Minor points:

1) Comparison among the K_d reported in Fig 1a,b with panel 1 d-f based on the ITC experiments; the experiments, as indicated in the figure, have been performed under different salt conditions, thus they are not comparable

2) Page 10 (about half page), the authors claim: "During interphase, we observed constitutive vesicle-Golgi fusion via p115 and GM130 interaction, as revealed by punctate fluorescence". I think that the conclusion "constitutive vesicle-Golgi fusion" is an indirect extrapolation of the data; I suggest to use a more general description.

3) Figure of wound healing without quantification does not add much. Either remove or quantify

Reviewer #3 (Remarks to the Author):

Golgi partitioning during mitosis is thought to depend on its vesicularization, driven at least in part by disruption of the interaction between GM130 and p115, two factors whose binding is required for Golgi fusion. Chang et al report that the transport factor importin α is a major regulator of the GM130-p115 interaction. The authors show that when the previously identified serine residue (Ser-35) of GM130 is phosphorylated, its binding affinity for importin increases, thereby outcompeting p115-GM130 binding. A key finding, provided by a high-resolution crystal structure of the binding interface between GM130 and importin α , highlights non-canonical binding through multiple contact sites in addition to the two bipartite NLS binding sites, which is further enhanced by Ser-35 phosphorylation. The authors also identify a lysine residue within GM130 (Lys-34) that when mutated to alanine blocks importin α binding. A CRISPR'd cell line harboring this mutation shows disrupted GM130-importin α binding and Golgi disassembly during mitosis appears aberrant. Overall the findings are significant and uncover a plausible model for Golgi disassembly during mitosis in a RanGTP-independent mechanism via GM130 binding importin α in a novel, non-canonical manner. The following points should be addressed:

1) Figure 1C: The authors show a titratable effect of importin α on the S25D mutant form of GM130 binding to p115. An important control would be to compare the wild-type GM130 protein in parallel, which should not be able to bind as well to importin α to compete for p115 binding.

2) In Figure 4d the authors use a GM130 antibody to demonstrate a change in Golgi dispersion upon GM130 K43A mutant expression by immunofluorescence. However, the images shown indicate that the Golgi is even more dispersed, when the prediction is that dispersion should be impaired. A distinct Golgi marker should be used instead of the GM130 antibody and the results quantified.

3) The authors use two different lysine mutants of GM130 to show that these mutants, which abrogate importin α binding, inhibit Golgi disassembly and affect the cell cycle. Since lysine residues are frequently ubiquitylated to promote protein degradation, protein stability of mutant and wildtype proteins should be compared by Western blot following cycloheximide treatment.

4) The model in Figure 2f is not a clear representation of the author's hypothesis. It should show that importin α binding of GM130 inhibits its interaction with p115.

Reviewers' comments:

Reviewer #1 (Remarks to the Author):

This is an interesting paper that describes a potentially novel function of importin alpha as a regulator of Golgi disassembly. Unfortunately, the paper is unusually concise and falls short in describing important experimental details. I have some general comments that the authors should consider and address in order to make this work better readable and scientifically sound.

Major points

1. Title. It is not clear to me why this process would be Ran-independent. As long as GM130 binds to both importin alpha and importin beta, RanGTP is still involved. I don't see any experiment in the paper that rules out this possibility.

We thank the reviewer for his/her comments. Conventionally, cargos and Importin- α / β form a ternary protein complex, and binding of cargos to Importin- α depends on RanGTP and Importin- β . Presence of RanGTP disassembles the cargo/Importin- α / β complex, thereby releasing the cargos. In this study, we found that GM130 carries an "IBB-like" NLS and could bind full-length Importin- α and β (Fig. 3 and 4). Binding of Importin- α to GM130 occurs in the absence of Importin- β and is enhanced by a phosphate moiety. Importantly, we further demonstrate that this interaction modulates mitotic Golgi disassembly. Hence, we concluded that the conventional Ran-Importin- β pathway does not regulate the Importin- α /GM130 interaction that controls Golgi disassembly.

GM130 carries an "IBB-like" NLS and interacts with Importin- β (Fig. 4). Furthermore, the IBB of Importin- α and Ran displace GM130 from this interaction (Fig. 4g, j, k), suggesting that the GM130 NLS-like motif structurally and functionally resembles an IBB domain. Notably, the Importin- β and GM130 interaction is repressed by Ser-25 phosphorylation of GM130 (Fig. 4e, f). Hence, Ran or the phosphate moiety dissociates GM130 from Importin- β but not Importin- α (Fig. 4e, f, j, k), explaining why Importin- α but not Importin- β associates with GM130 in membrane fractions from either *Xenopus* egg extracts or mitotic mammalian cells. To emphasize that the Ran pathway-independent (phosphorylation-enhanced) Importin- α /GM130 interaction is crucial for mitotic Golgi disassembly and mitotic progression, we have slightly modified the title to "**Ran pathway-independent regulation of mitotic Golgi disassembly by Importin- α** ". Moreover, as the reviewer pointed out that the GM130 and Importin- α interaction *per se* is resistant to IBB displacement and requested that we explain how GM130 interacts with full-length Importin- α , we have changed one of the section headings to "**The GM130 and Importin- α interaction is resistant to IBB displacement and is enhanced by GM130 phosphorylation**" and now explain these traits in detail in the main text. We apologize for the confusion.

2. The Introduction is skimpy and fails to reference important published work on membrane protein NLSs. In addition, it doesn't touch upon the broad field of

nuclear transport.

We thank the reviewer for these comments. We have expanded our Introduction section and now include paragraphs in which we introduce nuclear import and membrane protein NLS.

3. Crystallography. The Rwork/Rfree values for GM130-WT and GM130-S25D are unacceptably high (the Rfree) and uncoupled raising questions on the quality of the NLS model. I ask the authors to do three things: First, the authors refine their model and lower the Rfree by at least 5-7 points (at least);

We thank the reviewer for these suggestions. In response, we have collected a new dataset on the GM130-S25D•Importin- α complex at a resolution of 1.75 Å (Table 1) and performed another refinement of the GM130-WT•Importin- α complex. The Rfree values for the GM130-WT•Importin- α and GM130-S25D•Importin- α complexes are 25.1 and 19.9, respectively (Table 1). Although we could only reduce the Rfree value by about 3% for the GM130-WT•Importin- α complex (28% versus 25%) compared to the previous model, we believe this Rfree value is generally acceptable for a middle resolution structure (2.8 Å). Additionally, we have now determined a high-resolution S25D crystal structure (1.75 Å) and assessed the side-chain density and peptide chain connectivity in the high-resolution electron-density map, allowing us to unambiguously build the model, which confirmed the accuracy of our GM130-WT•Importin- α model.

Second, they should show an OMIT map for the NLS, especially the novel alpha helix feature;

We thank the reviewer for this suggestion and have now included omit maps for the NLS of the GM130-WT•Importin- α and GM130-S25D•Importin- α complexes (Fig. 1h and Extended Data Fig. 1c).

Third, in the crystallographic table (that should be in the main text) they should provide the B-factor for the data (Wilson b-factor), importin alpha model, NLS, and water.

We thank the reviewer for these suggestions. The crystallographic table has now been moved from the Supplemental Information into the main text (Table 1), and B-factors have been included. Additionally, we also provided a figure (Extended Data Fig. 2) and explanation showing why the B-factor of the central α -helix in the GM130 phospho-S25•Importin- α complex is higher than for other complexes (under the section “**Ser-25 phosphorylation of GM130 indirectly enhances Importin- α binding**”).

4. Phosphorylation at GM130-S25. It is not clear how the phosphate at position 25 of GM130 is stabilizing the NLS-helix. This should be shown in greater detail and substantiated by experimentally or computationally (e.g., Stability curves, Molecular Dynamics simulation, etc.)

We thank the reviewer for this comment. As suggested by the reviewer, we have conducted multiple molecular dynamic (MD) simulations of wild-type and phosphorylated GM130-NLS free in solution and in complex with Importin- α . The simulations suggest that destabilization of the phosphorylated peptide relative to wild-type in solution contributes to the enhanced affinity of phosphorylated GM130-NLS for Importin- α . Whereas the two-turn α -helix was maintained in MD simulations of the wild-type and phosphorylated GM130-NLS-Importin- α complexes, the C-terminal region was disrupted in simulations of apo GM130-NLS phospho-S25 (Fig 2f). When bound to Importin- α , the S25 phosphate enables water-mediated intramolecular interactions with backbone amides of nearby residues. Thus, S25 phosphorylation seems to destabilize the free peptide and stabilize the bound form, leading to a slightly higher binding affinity. Most classical NLS sequences lack secondary structural elements and are likely to be unstructured in proteins. However, the GM130 NLS contains an α -helix that may interfere with the GM130 and Importin- α interaction. Hence, phosphorylation on GM130 “shortens” this α -helix, promoting GM130 and Importin- α binding (Fig. 2g).

Additionally, we carried out thermal-dependent circular dichroism (CD) to determine the melting temperature (T_m) of protein complexes and examined if GM130 phosphorylation endows conformational stability on the GM130•Importin- α complex. The temperature scans for Importin- α in complex with GM130-WT, S25D or phospho-S25 showed that the phosphorylated protein complexes exhibited increased melting temperatures (Fig. 2e, Extended Data Fig. 1d-f). The T_m shifted from ~ 44 °C for the wild-type complex to ~ 46 °C for the phosphorylated protein complexes. Since the phosphorylated GM130•Importin- α complexes (S25D or phospho-S25) displayed a lower T_m (Fig. 2e), more inter-molecular hydrogen bonds (Extended Data Fig. 1a), and reduced B-factors (Extended Data Fig. 2a), we propose that phosphorylation of Ser-25 enhances the binding affinity of GM130 for Importin- α via an indirect effect, i.e., by blocking the p115 and GM130 interaction.

Based on these findings, we propose that phosphorylation plays a regulatory role in two ways: (1) by destabilizing the conformation of free GM130-NLS; and (2) by stabilizing the conformation of GM130-NLS bound to Importin- α .

5. GM130-NLS binding to importin beta. Does phosphorylation of S25 affect binding to importin beta? This should be shown in Figure 3d.

As suggested by the reviewer, we carried out ITC to examine whether phosphorylation on GM130 affects Importin- β binding. Interestingly, phosphopeptide (GM130-NLS phospho-S25) showed a binding affinity to Importin- β of at least three orders of magnitude lower than to the wild-type GM130 peptide (Fig. 4e, f). This result suggests that the phosphor moiety on GM130 substantially reduces the binding affinity for Importin- β . Notably, IBB uses an “unfolded” configuration to interact with Importin- α , but adopts a “folded” α -helix conformation in complex with Importin- β ²⁹. Hence, destabilization of the apo-GM130 peptide α -helix upon phosphorylation revealed by MD simulations may cause the reduction of binding affinity between GM130 and Importin- β (Fig. 2f).

6. GM130-NLS competition with IBB. Carry out a competition assay between GM130-NLS and IBB-domain to determine if binding to importin b is mutually exclusive. Also, in reference to point #1, why would the process Ran-independent?

We thank the reviewer for this suggestion and comment. We carried out competition assays and found that GM130-S25D was able to pull down Importin- β and, moreover, it could be displaced by IBB in this interaction (Fig. 4g). Hence, the binding site on Importin- β for GM130 and IBB is mutually exclusive. While the Importin- β /GM130 interaction is Ran-dependent, we have demonstrated that the Importin- α /GM130 interaction is Ran/Importin- β -independent and is crucial for mitotic Golgi disassembly and mitotic progression. Thus, to avoid confusion, we have modified the title to “**Ran pathway-independent regulation of mitotic Golgi disassembly by Importin- α** ”.

7. Affinity for FL-Importin alpha. The affinity of GM130-NLS for importin beta is 1.2 μ M, and there's some ~3 μ M importin b in a cell, which means GM130 will be mostly bound to importin beta at steady state. In contrast, the measured affinity of GM130-NLS for FL-importin alpha is 6.9 μ M (2.1 μ M after phosphorylation), but the estimated concentration of importin alpha is ~1 μ M. Thus, what is the biological significance of focusing this paper on a 'novel' property of importin alpha which is not likely to be significant in vivo?

We thank the reviewer for these observations. Both the Ran pathway and GM130 phosphorylation suppress the GM130 and Importin- β interaction (Fig. 4e, f, j, k). However, binding of GM130 to full-length Importin- α is resistant to IBB displacement and is enhanced by GM130 phosphorylation (Fig. 3). These findings explain why sub-stoichiometries of Importin- α and - β have been associated with *Xenopus* egg membrane fractions and can be pulled down by GM130 from mitotic cell lysates, as demonstrated by other research groups.

Moreover, GM130 has been shown to function as a “membrane-associated” “higher-order” structural assembly *in vivo* (e.g. tetramers), so a higher local concentration of GM130 may facilitate the GM130 and Importin- α interaction, despite both the biochemical binding affinity of phosphorylated GM130-NLS to full-length Importin- α and the cellular concentration of Importin- α being comparable (i.e., in the sub-micromolar range). Additionally, our cell lysate pull-down experiments showed sub-stoichiometric pull-down of Importin- α and - β (Extended Data Fig. 1e-k). Importantly, Importin- α pull-down by GM130 could be reduced by the K34A mutation of GM130 in cell lysates (Fig. 5b, 6c), resulting in impaired mitotic Golgi disassembly. Hence, these findings support the biological significance of the Importin- α and GM130 interaction.

8. Role of Ran. Section “Conventionally, GTP-bound Ran binds to importin b....” There seems to be a misunderstanding in the way the authors describe the putative role of Ran on importin a:Gp130-NLS interaction. The authors refer to this interaction as Ran-independent and even the title hints at this. However, NLS-binding to importin a is always Ran-Independent. The authors are confusing

RanGTP-mediated displacement of importin alpha:NLS from importin alpha with the auto-displacement of the NLS from importin alpha that is mediated by the IBB. What the authors need to demonstrate here is that the GM130-NLS is immune from IBB-autoinhibition, and that is not likely to be the case given the Kd measured by ITC. The authors should set up a pull-down like in Fig 1c that has as control DIBB-importin alpha. The quantity of NLS-cargo pulled-down by FL-importin alpha has to be normalized to the total GM130-NLS captured by the DIBB-importin alpha.

We thank the reviewer for these comments and apologize for the confusion. We carried out a competition assay and found that IBB was essentially not able to displace GM130 in the GM130•Importin- α (Δ IBB) complex at an 8-fold higher concentration (Extended Data Fig. 3i). Together with our ITC data (Fig. 1c, Extended Data Fig. 3h), we conclude that the GM130-NLS is an IBB-like NLS and is immune from IBB-autoinhibition, as pointed out by the reviewer. Furthermore, to avoid confusion, we have modified the main text and changed the section heading to “**The GM130 and Importin- α interaction is resistant to IBB displacement and is enhanced by GM130 phosphorylation**” and altered the title to “**Ran pathway-independent regulation of mitotic Golgi disassembly by Importin- α** ”.

Additionally, as per the reviewer’s suggestions, we carried out binding assays using GST-IBB (control), GST-GM130-S25D and GST-GM130-S25D/K34A to pull down either full-length Importin- α or Importin- α (Δ IBB). Based on our “side-by-side” binding assays, we can reveal that GM130-S25D was able to bypass IBB auto-inhibition, as ~50% of Importin- α FL was pulled down by GM130-S25D relative to Importin- α (Δ IBB), which was substantially more than pulled down by the control (GST-IBB) (Extended Data Fig. 4b, c). Importantly, the ability of the K34A mutant to overcome IBB displacement was greatly reduced, i.e. to an extent even lower than for the control (Extended Data Fig. 4b, c). Hence, we used this mutant to carry out subsequent cell biology-based experiments and to examine the function of the GM130 and Importin- α interaction.

9. Figure 3f. Repeat the RanQ69L displacement pull-down in the presence of physiological concentrations of importin beta (~3 μ M) and FL-importin alpha (~1 μ M).

Fig. 1 for reviewer: GST pull-down assays of GM130-S25D with Importin- α FL (1 μ M) or Importin- β (3 μ M) in the presence or absence of RanQ69L.

We thank the reviewer for this suggestion. We have now carried out competition pull-down assays using 3 μ M of Importin- β and 1 μ M of Importin- α FL (Fig. 1 for reviewer). Since these concentrations of Importin- α and - β are lower than those we used in Fig. 4j (5 μ M), pull-

down of both proteins by GST-GM130 was reduced based on band intensities. However, our finding that Importin- β is displaced by RanQ69L remains the same, so we have retained the original results/figures in the revised manuscript (Fig. 4j).

GM130 functions as a “membrane-associated” “higher-order” structural assembly *in vivo* (e.g. tetramers), so a higher local concentration of GM130 may facilitate the GM130 and Importin- α interaction, despite both the biochemical binding affinity of phosphorylated GM130-NLS to full-length Importin- α and the cellular concentration of Importin- α being comparable (i.e. in the sub-micromolar range). Additionally, given the presence of abundant cargo substrates in the cell, it is very difficult to estimate “free” Importin- α and Importin- β under cellular conditions. Hence, we have verified our biochemical findings by means of cell lysate pull-down and cell-based assays.

Overall, this paper is not presentable in the current form but needs additional experiments and controls.

Reviewer #2 (Remarks to the Author):

The paper by Chih-Chia Chang et. reveals a novel mechanistical aspect of the regulation of mitotic Golgi disassembly. In particular, it has been known for more than two decades that Golgi disassembly is also induced by inhibition of membrane fusion in consequence of suppression of the interaction of the Golgi matrix proteins p115 and GM130. This inhibition correlates with mitotic phosphorylation of GM130. However, the mechanism through which phosphorylation affects the binding of the proteins, and the physiological consequences of this inhibition, were not known.

Here Chang and co-workers show the unexpected and interesting finding that inhibition p115/GM130 interaction is mediated by the binding of Importin- α (a nuclear transport factor) to GM130. By using a series of biochemical approaches, the authors show that Importin- α directly competes with p115 for the binding to GM130. The binding of Importin- α is promoted by phosphorylation of Ser25 of GM130. By co-crystallization and modelling experiments, the authors identify the minimal and essential requirements for Importin- α /GM130 interaction. Based on this, they design GM130 point-mutants that are not able to bind Importin- α , and they show that cells expressing the mutant displayed large Golgi aggregates during mitosis and mitosis defects, in line with a less effective disruption of the interaction of p115 with GM130. In general, I think that the paper is interesting and reveals a novel mechanism of regulation of Golgi fragmentation during mitosis, and they also suggest an intriguing possible coordination with nuclear disassembly. I think that the general conclusion of the paper are plausible and should be of interest to a wide audience. In general, the paper is technically sound, but it is missing a series of important controls and quantifications. Moreover, I suggest that a deeper investigation of the physiological consequences of alterations of the binding of importin to GM130 would strengthen the paper. Thus, I suggest that the following points should be addressed before publication.

Major concerns:

1) The conclusions based on the pull-down or co-IP experiments appear plausible and are in line with the other approaches. Yet, the experiments are missing important controls. The western blots of Fig 3c, 4h and extended 2e should also show the amount of the proteins of interest before (“input”) and after (“unbound”) the pull-down or IP. Moreover, the results should be expressed in a quantitative manner; this is particularly important when comparing the differential co-IP of importin- α and importin- β with GM130. In addition, as the in vitro interaction of importin- α with GM130 appears to be regulated by phosphorylation (most probably by Cdk1), it would be interesting to test the effect of Cdk1 inhibition on the interaction of the endogenous proteins.

We thank the reviewer for his/her comments. In response, we have now included input and unbound controls for all Western blots (Extended Data Fig. 3e, Fig. 5b and Fig. 6c). In addition to now showing pull-down or IP results in a quantitative manner for cell lysate pull-down experiments using recombinant GM130 (Extended Data Fig. 3i, Fig. 5b), we also present quantitative plots of input and unbound band intensities.

Even though we incubated excess GST-GM130 recombinant protein, only small fractions of Importin- α and - β were pulled down by GM130 (Extended Data Fig. 3e), probably due to the presence of abundant NLS-containing proteins in the cell lysate. However, Importin- α pull-down by GM130-S25D was elevated ~ 1.3 -fold compared to WT (Extended Data Fig. 3g), suggesting that phosphorylated GM130 has a better affinity for Importin- α than wild-type, even in the presence of other NLS proteins. Additionally, though we detected about the same protein quantities of Importin- α and - β in the cell lysates (Extended Data Fig. 3f), GM130 pulled down ~ 10 -fold more Importin- α than Importin- β (Extended Data Fig. 3g). These observations essentially recapitulate the sub-stoichiometric ratio of Importin- α and Importin- β associated with membrane fractions of *Xenopus* egg extracts and that is pulled down by GM130 from HeLa cell lysates.

To examine whether inhibition of Cdk1 activity could alter the binding of Importin- α and p115 to GM130, we treated nocodazole-arrested cells with the Cdk1 inhibitor RO-3306 (Vassilev et al., *PNAS* 2006) and then conducted IP analysis on RO-3306-treated or -untreated cell lysates using GM130 antibody. We used a commercially-available antibody that detects Ser-25 phosphorylation (Santa Cruz: sc-377549) to examine the efficiency of RO-3306 to inhibit Cdk1 activity. However, the GM130 antibody we used did not recognize mitotic phosphorylated GM130. Hence, as Vassilev et al. had shown that 1-hour RO-3306 treatment was sufficient to block phosphorylation of most Cdk1 substrates and induced a change in cell morphology from a rounded mitotic cell to a flat interphase shape, we assessed whether Cdk1 inhibits GM130 phosphorylation based on changes in cell shape (Extended Data Fig. 5b). Similar to what we observed for the K34A mutant (Fig. 6c, d), we found that the amounts of Importin- α and p115 pulled down by GM130 were reduced or enhanced, respectively, in RO-3306-treated cell lysates isolated from cells with a flat morphology (Extended Data Fig. 5c, d). Thus, we have demonstrated that fragmentation of the Golgi apparatus into vesicles, which is crucial for mitotic progress, is modulated by Importin- α -GM130 interaction, which is facilitated by Cdk1-mediated GM130 phosphorylation.

2) The investigation of the physiological consequences of the GM130 mutation appears to be limited; one concern is related to the use of the few heterozygous colonies survived during the CRISPR/cas9 genome editing procedure. I think that a similar set of experiments should be performed in cells depleted by siRNA and/or expressing the GM130 mutants that do not bind to importin- α ; as a readout, the authors should examine Golgi fragmentation (in a quantitative manner) and mitotic defects. In addition, I think that it is important to try to address the core mechanism for the mitotic defect. For example, one possible cause is the reduced level of Golgi fragmentation; the other could be that the GM130 mutant remains bound to p115 and, as a consequence, does not work properly in stabilizing the mitotic spindle (Wei et al J Cell. 2015). Have the authors tested if the cells expressing GM130 mutant show spindle defects?. I think that the authors should elaborate more, in terms of experimenst and discussion, on this aspect.

We thank the reviewer for these suggestions. In response, we investigated whether the impaired Golgi disassembly resulting from reduced Importin- α and GM130 interaction could cause mitotic spindle defects. The K34A mutant cell lines displayed a higher percentage of mitotic cells compared to that of wild-type (Fig. 6e), mirroring the higher proportion of G2/M cells in the mutant cell lines upon quantitation of DNA content (Fig. 5e, f). Moreover, the mutant cells showed a 3- to 4-fold increase in the percentage of mitotic cells with monopolar spindles relative to wild-type (Fig. 6h, i), suggesting that the K34A mutation on GM130 reduces GM130 and Importin- α interaction and promotes monopolar spindle formation during mitosis.

Overexpression of the K34A mutant using the CMV promoter and depletion of endogenous GM130 by small interfering RNA (siRNA) also resulted in elevation of the mitotic cell percentage compared to controls (Fig. 6e-g). Overexpression and knockdown efficiencies of GM130 were assessed by Western blot analyses (Extended Data Fig. 6a, b). Additionally, we carried out immunofluorescence staining to monitor whether Golgi-bound GM130 is depleted, since siRNA oligonucleotides partially knocked down GM130 (Extended Data Fig. 6b). Upon GM130 depletion with 50 or 100 nM siRNA, GM130 fluorescence on Golgi was barely detectable (Extended Data Fig. 6c). Moreover, similar to observations reported by Kodani and Sutterlin (*Mol Biol Cell* 2008), GM130 depletion resulted in mislocalization at Golgi of GRASP65 but not Giantin (Extended Data Fig. 6c-e), suggesting that Golgi-bound GM130 is substantially depleted.

Next, we assessed all mitotic cells to establish how many had monopolar or multiple polar spindles (Fig. 6h). When GM130 was overexpressed (K34A mutant) or depleted, there was an equivalent ratio of monopolar and multiple spindles in the mitotic cells (Fig. 6k), even though the proportion of cells having monopolar spindles was slightly increased in the two concentrations of siRNA that mediated GM130 depletion (50 and 100 nM, Fig. 6j). These results suggest that knockdown of GM130 concurrently increases the numbers of mitotic cells containing monopolar and multiple polar spindles. Notably, in the K34A mutant cell lines, mitotic cells with monopolar spindles were approximately 2- to 3-fold more abundant than cells with multiple polar spindles (Fig. 6k). Therefore, our findings imply that the monopolar spindle defect most likely is specifically enhanced by the point mutation on GM130. Notably, chemically-inhibited Golgi disassembly has been shown to arrest cells with monopolar spindles, as demonstrated by Guizzunti and Seemann (*PNAS* 2016). Hence, we conclude that Importin- α -GM130-facilitated mitotic Golgi disassembly modulates spindle assembly and mitotic progression. Additionally, as GM130 has multiple important cellular functions in interphase and mitosis, other pathways through indirect or combined effects may cause the mitotic defect (i.e. the high percentage of mitotic cells) resulting from GM130 overexpression and depletion.

Taken together, our findings show that GM130 phosphorylation (carried out by Cdk1) enhances binding of Importin- α released from SAFs by Ran, thereby preventing p115 binding to GM130 and promoting Golgi disassembly. Golgi disassembly further ensures proper spindle assembly. Hence, Importin- α coordinates the Ran and kinase-phosphatase pathways to regulate major events during mitosis (NEBD, Golgi disassembly and spindle assembly) (Fig. 7b).

3) Elaborate and discuss more on the model. For example, is the binding of

Importin- α to GM130 correlated with nuclear envelope breakdown?. This could suggest a functional coordination between nuclear envelope breakdown and disassembly of the Golgi stacks and formation of the spindle.

We thank the reviewer for this suggestion. We have now modified the main text/figures, and have included the following paragraph in the Discussion section:

The nuclear envelope undergoes breakdown (NEBD) at the onset of open mitosis, requiring phosphorylation of numerous proteins residing in the nuclear envelope (e.g. nuclear lamins) by cyclin-dependent kinase 1 (Cdk1). Notably, Cdk1 also phosphorylates the Ser-25 residue within the NLS-like motif of GM130 during early mitosis, and this GM130 phosphorylation event has been correlated with mitotic Golgi disassembly. Here, we have demonstrated that phosphorylation of Ser-25 enhances the GM130 and Importin- α interaction, thereby sterically blocking the p115-binding site of GM130. During early mitosis, RanGTP releases Importin- α / β from spindle assembly factors (SAFs) (e.g. TPX2 and NuMA), activating the mitotic functions of these SAFs and in turn promoting mitotic spindle assembly. Thus, the Importin- α liberated by RanGTP can instead inhibit the p115-binding function of Cdk1-phosphorylated GM130, suppressing p115-mediated vesicle-Golgi fusion and allowing Golgi disassembly. Taken together, our findings provide a model whereby the Ran and kinase-phosphatase pathways coordinately regulate major events during mitosis (NEBD, Golgi disassembly and spindle assembly) to ensure mitotic progression, facilitated by Cdk1 and the nuclear transport factor Importin- α (Fig. 7b).

Minor

points:

1) Comparison among the Kd reported in Fig 1a,b with panel 1 d-f based on the ITC experiments; the experiments, as indicated in the figure, have been performed under different salt conditions, thus they are not comparable

We thank the reviewer for this observation and apologize for the confusion. As pointed out by the reviewer, the ITC experimental conditions for p115CTD/GM130 (Fig. 1a, b) and Importin- α (Δ IBB)/GM130 (Fig. 1d-f) contained 50 and 200 mM NaCl, respectively. Since binding affinity between proteins is typically reduced if a higher concentration of salt is applied in ITC, we have modified the sentence to emphasize that Importin- α (Δ IBB)/GM130 showed a greater binding affinity compared to p115CTD/GM130 even under higher salt conditions (~126 nM in 200 mM NaCl versus ~980 nM in 50 mM NaCl). Importantly, our competitive binding assays further showed that both Importin- α (Δ IBB) (Fig. 1c) and full-length Importin- α (Fig. 3d) could compete with p115-CTD for GM130 binding.

2) Page 10 (about half page), the authors claim: “During interphase, we observed constitutive vesicle–Golgi fusion via p115 and GM130 interaction, as revealed by punctate fluorescence”. I think that the conclusion “constitutive vesicle–Golgi fusion” is an indirect extrapolation of the data; I suggest to use a more general description.

We thank the reviewer for this comment. We have modified the sentence to “we observed punctate fluorescence concentrated near the nuclei of both wild-type and mutant cell lines, indicating that the vesicle–Golgi fusion facilitated by p115 and GM130 interaction is comparable in wild-type and mutant lines (Fig. 6a, interphase cells)”.

***3) Figure of wound healing without quantification does not add much. Either
remove or quantify***

We thank the reviewer for this suggestion and have removed the wound-healing results from the revised manuscript.

Reviewer #3 (Remarks to the Author):

Golgi partitioning during mitosis is thought to depend on its vesicularization, driven at least in part by disruption of the interaction between GM130 and p115, two factors whose binding is required for Golgi fusion. Chang et al report that the transport factor importin alpha is a major regulator of the GM130-p115 interaction. The authors show that when the previously identified serine residue (Ser-35) of GM130 is phosphorylated, its binding affinity for importin increases, thereby outcompeting p115-GM130 binding. A key finding, provided by a high-resolution crystal structure of the binding interface between GM130 and importin α , highlights non-canonical binding through multiple contact sites in addition to the two bipartite NLS binding sites, which is further enhanced by Ser-35 phosphorylation. The authors also identify a lysine residue within GM130 (Lys-34) that when mutated to alanine blocks importin α binding. A CRISPR'd cell line harboring this mutation shows disrupted GM130-importin α binding and Golgi disassembly during mitosis appears aberrant. Overall the findings are significant and uncover a plausible model for Golgi disassembly during mitosis in a RanGTP-independent mechanism via GM130 binding importin α in a novel, non-canonical manner. The following points should be addressed:

1) Figure 1C: The authors show a titratable effect of importin α on the S25D mutant form of GM130 binding to p115. An important control would be to compare the wild-type GM130 protein in parallel, which should not be able to bind as well to importin α to compete for p115 binding.

We thank the reviewer for his/her suggestions. Since full-length Importin- α (FL) is more physiologically relevant, we carried out parallel competitive binding assays using Importin- α FL and p115-CTD in the presence of either GM130-WT or GM130-S25D. As for Importin- α (Δ IBB) (Fig. 1c), Importin- α FL also competed with p115-CTD for GM130 binding (Fig. 3d). Importantly, the p115-CTD•GM130-S25D complex was more sensitive to the competition by Importin- α FL compared to p115-CTD•GM130-WT (Fig. 3d-f). This result suggests mitotic GM130 phosphorylation enhances Importin- α binding, promoting dissociation of p115.

2) In Figure 4d the authors use a GM130 antibody to demonstrate a change in Golgi dispersion upon GM130 K43A mutant expression by immunofluorescence. However, the images shown indicate that the Golgi is even more dispersed, when the prediction is that dispersion should be impaired. A distinct Golgi marker should be used instead of the GM130 antibody and the results quantified.

We thank the reviewer for this suggestion and apologize for the confusion. We have modified the sentence to: “The K34A mutant cell lines presented a **scattered, punctate pattern** of greater fluorescence intensity during metaphase compared to the **more evenly dispersed** GM130 signal in wild-type metaphase cells (Fig. 5g, h, Extended Data Fig. 5a)”.

Additionally, we used two Golgi markers (GRASP65 and Giantin) to further validate this finding. In the K34A mutant, metaphase cells stained by GRASP65 and Giantin

antibodies displayed comparably elevated fluorescence intensities and punctate fluorescence patterns, as observed by GM130 antibody-staining in the mutant (Fig. 5g-l).

3) The authors use two different lysine mutants of GM130 to show that these mutants, which abrogate importin α binding, inhibit Golgi disassembly and affect the cell cycle. Since lysine residues are frequently ubiquitylated to promote protein degradation, protein stability of mutant and wildtype proteins should be compared by Western blot following cycloheximide treatment.

We thank the reviewer for this suggestion. We treated cell lines that expressed wild-type GM130 or the K34A mutant with the protein synthesis inhibitor cycloheximide to measure protein stability. Both wild-type and K34A mutant proteins were equally stable in cells, as indicated by the comparable band intensities in Western blots between these two proteins over a time-course (Extended Data Fig. 4d).

4) The model in Figure 2f is not a clear representation of the author's hypothesis. It should show that importin α binding of GM130 inhibits its interaction with p115.

We thank the reviewer for this observation and have modified the model accordingly (Fig. 2g and 7a).

REVIEWERS' COMMENTS:

Reviewer #1 (Remarks to the Author):

I am satisfied by the way the authors took all my suggestions seriously and worked hard to carry out additional experiments which greatly expand the breath and depth of this paper.

I only have two minor suggestions:

1. Title: Though it is not my call to pick a title for this paper, I find "Ran pathway-independent regulation of mitotic Golgi disassembly by Importin- α " to be missing the point of this paper. I wonder if a more vanilla "Regulation of mitotic Golgi disassembly by Importin- α ", or something along these lines, would be better suited.

2. Line 4, page 5: "To prove our hypothesis". Please revise "To Test our hypothesis". We really don't do science to prove hypothesis but to test them.

Overall, I think this paper is now ready for publication in NC.

Gino Cingolani, Ph.D.

Reviewer #2 (Remarks to the Author):

I am satisfied with the authors' rebuttal. I think that the paper is interesting and reveals a novel mechanism of regulation of Golgi fragmentation during mitosis. Moreover, in its current form, the paper shows interesting data about the physiological consequences of the GM130 mutation.

Reviewer #3 (Remarks to the Author):

The authors have addressed all of the concerns and the manuscript is now highly suitable for publication in Nature Communications.

Reviewer #1 (Remarks to the Author):

I am satisfied by the way the authors took all my suggestions seriously and worked hard to carry out additional experiments which greatly expand the breath and depth of this paper.

I only have two minor suggestions:

1. Title: Though it is not my call to pick a title for this paper, I find “Ran pathway-independent regulation of mitotic Golgi disassembly by Importin- α ” to be missing the point of this paper. I wonder if a more vanilla “Regulation of mitotic Golgi disassembly by Importin- α ”, or something along these lines, would be better suited.

We appreciate the reviewer’s comments. In this study, we demonstrate that mitotic Golgi disassembly is modulated by Importin- α -GM130 interaction, which is provoked by a Ran-independent pathway. The original title “**Ran pathway-independent regulation of mitotic Golgi disassembly by Importin- α** ” therefore more accurately reflects the contents of this manuscript. We wish to emphasize the Ran pathway independence of this mechanism and thus, respectfully, would prefer to retain our original title.

2. Line 4, page 5: "To prove our hypothesis". Please revise "To Test our hypothesis". We really don't do science to prove hypothesis but to test them.

We thank the reviewer for this suggestion. We have made the change accordingly.

Overall, I think this paper is now ready for publication in NC.

We thank the reviewer for his comments and suggestions that have allowed us to improve the manuscript.

Gino Cingolani, Ph.D.

Reviewer #2 (Remarks to the Author):

I am satisfied with the authors' rebuttal. I think that the paper is interesting and reveals a novel mechanism of regulation of Golgi fragmentation during mitosis. Moreover, in its current form, the paper shows interesting data about the e physiological consequences of the GM130 mutation.

We thank the reviewer for his/her previous comments and suggestions that allowed us to improve the manuscript.

Reviewer #3 (Remarks to the Author):

The authors have addressed all of the concerns and the manuscript is now highly suitable for publication in Nature Communications.

We thank the reviewer for his/her previous comments and suggestions that allowed us to improve the manuscript.